# Biosensor Guided Polyketide Synthases Engineering for Optimization of Domain Exchange Boundaries

Elias Englund [1,2], Matthias Schmidt [1,3,4], Alberto A. Nava [1,5], Sarah Klass [1,4,5], Leah Keiser [1,5], Qingyun Dan[1,4], Leonard Katz[1,6], Satoshi Yuzawa[1,4,7,8] & Jay D. Keasling[1,4,5,6,9,10,11] ✉

Type I modular polyketide synthases (PKSs) are multi-domain enzymes functioning like assembly lines. Many engineering attempts have been made for the last three decades to replace, delete and insert new functional domains into PKSs to produce novel molecules. However, inserting heterologous domains often destabilize PKSs, causing loss of activity and protein misfolding. To address this challenge, here we develop a fluorescence-based solubility biosensor that can quickly identify engineered PKSs variants with minimal structural disruptions. Using this biosensor, we screen a library of acyl-transferase (AT)-exchanged PKS hybrids with randomly assigned domain boundaries, and we identify variants that maintain wild type production levels. We then probe each position in the AT linker region to determine how domain boundaries influence structural integrity and identify a set of optimized domain boundaries. Overall, we have successfully developed an experimentally validated, high-throughput method for making hybrid PKSs that produce novel molecules.

Type I modular polyketide synthases (PKSs) are large, assembly line-like enzymes that are capable of producing structurally complex natural products, including many important pharmaceuticals with antibiotic, antiparasitic, and immunosuppressive properties[1,2]. A modular PKS is composed of several modules with each carrying out a single polyketide chain-extension and modification. Each module contains several catalytic domains. A ketosynthase (KS) catalyzes a decarboxylative Claisen condensation between a growing polyketide chain and an extension unit loaded by an acyltransferase (AT) onto an acyl carrier protein (ACP). In addition to the three core domains, a ketoreductase

(KR) domain, a dehydratase (DH) domain, and/or an enoyl reductase (ER) domain may be included in a module to reduce the b-keto group generated by chain elongation to a hydroxyl group, a carbon-carbon (C-C) double bond, or a saturated C-C bond. The full-length polyketide is lastly offloaded from the assembly line by a thioesterase (TE) domain.

Since the first gene cluster was discovered and the reaction mechanisms were proposed in the early 1990s, one of the most important goals in the field of PKS research is highly accurate rational protein engineering for generating novel molecules[3–5]. The modular

[1]Joint BioEnergy Institute, Emeryville, CA, USA. [2]School of Engineering Sciences in Chemistry, Biotechnology and Health, Science for Life Laboratory, KTH - Royal Institute of Technology, Stockholm, Sweden. [3]Institute of Applied Microbiology, Aachen Biology and Biotechnology (ABBt), RWTH Aachen University, Aachen, Germany. [4]Biological Systems and Engineering Division, Lawrence Berkeley National laboratory, Berkeley, CA, USA. [5]Department of Chemical & Biomolecular Engineering, University of California, Berkeley, Berkeley, CA, USA. [6]QB3, University of California, Berkeley, Berkeley, CA, USA. [7]Institute for Advanced Biosciences, Keio University, Tsuruoka, Yamagata, Japan. [8]Graduate school of Media and Governance, Keio University, Fujisawa, Kanagawa, Japan. [9]Department of Bioengineering, University of California, Berkeley, Berkeley, CA, USA. [10]Center for Biosustainability, Danish Technical University, Lyngby, Denmark. [11]Center for Synthetic biochemistry, Institute for Synthetic biology, Shenzhen Institute of Advanced Technology, Shenzhen, China. ✉e-mail: jdkeasling@lbl.gov

organization of the enzyme enables exchanging domains and modules to generate hybrid PKSs. Early studies demonstrated that combinatorial engineering of PKSs can generate new predicted polyketide structures, though with significantly reduced activities in most cases[6,7]. Since then, more precise protein engineering strategies have become the main thrust of the research to improve the activities of the hybrid PKS[8]. One of the most frequent targets for domain engineering has been the AT domain due to it being responsible for substrate incorporation in polyketide biosynthesis[9]. Precise AT engineering enables the diversification of functional groups (linear[6] and branched chains[10], terminal alkynes[11–13], phenyl groups[14], and halogenated alkyl groups[15]) attached to the polyketide backbone, and the production of target molecules with not only a high success rate but also an increased titer.

Although there are many examples of AT domain exchange to date, the resulting hybrid PKSs often show significantly reduced catalytic activities compared to their wild-type counterparts[6,8]. One of the major barriers to generating functional and stable hybrid PKSs is a poor understanding of where in the interdomain linkers one domain ends and another starts. Recently, the first crystal structure of an entire PKS module containing a KS, an AT, a KR, and an ACP (module 7 of the lasalocid PKS) has been reported[16]. Using this structural information, as well as other PKS module structures proposed by cryo-electron microscopy analysis[17,18], it should be possible to roughly obtain domain boundary information. However, to unambiguously predict domain boundaries, we believe that experimental validation of domain exchanged PKS libraries is still necessary.

To date, only three examples of randomized mutagenesis libraries of engineered PKSs have been reported. Two of them used DNA shuffling and yeast homologous recombination to generate libraries of pikromycin and the erythromycin PKS hybrids[19,20]. The third example serendipitously generated rapamycin analogs by recombination using plasmids with high sequence similarities in a *Streptomyces* strain[21]. Yet, these experiments still require screening of metabolite production from thousands of mutants to find a handful of active proteins[20].

Here we describe a method for efficiently sorting through a PKS library engineered with randomized linker junctions using a solubility biosensor in *Escherichia coli*. Using a previously described solubility biosensor[22,23], we screen a library of AT-exchanged variants to find those with high solubility and which are thus less likely to have disruptions to protein structure and function. To our knowledge, this is the first high-throughput screen based on protein stability that has been developed for PKSs.

## Results

### Development of solubility biosensor for PKSs

In our previous AT domain exchange effort, we observed a correlation between solubilities and in vitro activities of hybrid PKSs[8]. We reasoned that engineered PKSs that maintain a stable conformation, thereby avoiding aggregation, have a higher probability of exhibiting expected activities. To test that hypothesis, we sought to develop *E. coli* biosensor strains that could detect protein misfolding.

Several methods have already been developed for assaying in vivo protein stability with fluorometric outputs[22,24–26]. However, these methods have not been tested with large, multi-domain proteins such as PKSs. Heat-shock genes *ibpA* and *fxs* are highly expressed when misfolded proteins accumulate inside *E. coli*[27]. The promoters of those genes ($P_{ibp}$[22,28] and $P_{fxs}$[23]) were used to drive expression of the green fluorescent protein (*gfp*) gene ($P_{ibp}$ alone or $P_{ibp}$ and $P_{fxs}$ in tandem) and integrated into the genome of *E. coli* BL21 (DE3) in the *arsB* gene, thereby creating $\Delta arsB$::$P_{ibp}$ GFP and $\Delta arsB$::$P_{ibpfxs}$ GFP. The *arsB* site encodes an arsenic membrane pump, which should be a neutral site under standard laboratory conditions. In parallel, we made $\Delta ibpA$::GFP by integrating the *gfp* gene in frame with the *ibpA* gene, consequently knocking out the native gene and appropriating the promoter. To assess how these misfolded protein biosensors react to PKSs with

different levels of solubility, we used the sixth module of the erythromycin PKS (6-DeoxyErythronolide B Synthase) with the neighboring TE domain (DEBSM6) and two previously engineered versions, D0 and D1, that contained an AT from the epothilone PKS module 4 (EpoM4) in place of the native AT. DEBSM6 was previously shown to be more soluble than D1, which in turn was more soluble than D0 due to differences in linker junctions[8], i.e. the positions in the linker regions where the parental DEBS sequence ends and the heterologous AT sequence begins. As the AT domain is typically located in the middle of the PKS gene, PKSs engineered with AT exchanges carry two junctions, one upstream of the AT in the KS-AT linker and one downstream in the post-AT linker.

The three PKS variants together with an empty vector control were expressed from pET plasmids in the *E. coli* strains, and fluorescence was measured using a microplate spectrophotometer (Fig. 1a). The *ibpA* promoter integrated in the *arsB* locus was not induced by the highly soluble DEBSM6 while it was similarly induced by D0 and D1. Meanwhile, the tandem promoter $P_{ibpfxs}$ was more sensitive and became induced by DEBSM6 but to a lower degree than the other two PKSs. Integration of *gfp* into the *ibpA* locus gave high background GFP fluorescence with the empty vector, but it was lower than that of DEBSM6, which was lower than both D0 and D1. Of the three tested strains, $\Delta arsB$::$P_{ibp}$ GFP showed the most promising features as it had low leakiness and almost no activation by the most soluble PKS. Therefore, it was selected for later use.

Next, we investigated the use of fluorescent fusion tags to normalize biosensor activation to the amount of heterologous protein produced. Although other methods for monitoring PKS expression have been reported[29], the use of fusion tags is unexplored. Previous results from non-PKS proteins showed that the folding of a fluorescent fusion protein is affected by the folded state of the protein to which it is attached[24]. To evaluate the response of fusion tags with multidomain proteins like PKSs, mCherry was attached to the C-terminus of the three reference PKSs, thereby creating DEBSM6 mCherry, D0 mCherry, and D1 mCherry. The tagged PKSs were expressed, and the results showed lower fluorescence from D0 and D1 compared to DEBSM6 (Fig. 1b), which would be expected if mCherry has reduced fluorescence when attached to a less soluble protein. However, when we measured cellular abundance of the expressed PKSs by SDS-PAGE quantification, we found that the higher fluorescence in DEBSM6 mCherry was due to higher amounts of protein and not due to differences in solubility (Fig. 1b). To investigate why the less soluble D0 had proportionally the same fluorescence as DEBSM6, we separated soluble and insoluble fractions and measured the amounts of expressed PKSs using SDS-PAGE and fluorescence (Fig. 1c, d). We observed that mCherry fused to DEBSM6 did not significantly change the solubility, which was still more than 95% soluble, while D0 mCherry had more than half of its protein content in the insoluble fraction and D1 mCherry was 75% soluble. The fluorescence in the different fractions mirrored the protein content, indicating that the chromophore is still fluorescing even when attached to an insoluble protein. These results show no reduction in fluorescence depending on PKS solubility, in contrast to what has previously been observed for smaller proteins[25,30]. Therefore, we were unable to use the fused mCherry to indicate solubility, and it was instead used only as a measurement of protein expression.

To investigate biosensor activation at varying levels of expressed proteins, we combined the mCherry tagged PKSs with the *E. coli* strain harboring the $\Delta arsB$::$P_{ibp}$ GFP biosensor and induced protein expression by adding different concentrations of IPTG. Even at high IPTG concentrations (0.25 – 1 mM), DEBSM6 mCherry only weakly induced the biosensor, while D0 mCherry had high induction even at 50 μM (Fig. 1e). Furthermore, D0 mCherry exhibited higher GFP fluorescence than D1 mCherry, demonstrating that the biosensor can discriminate between the two proteins at appropriate IPTG concentrations. To

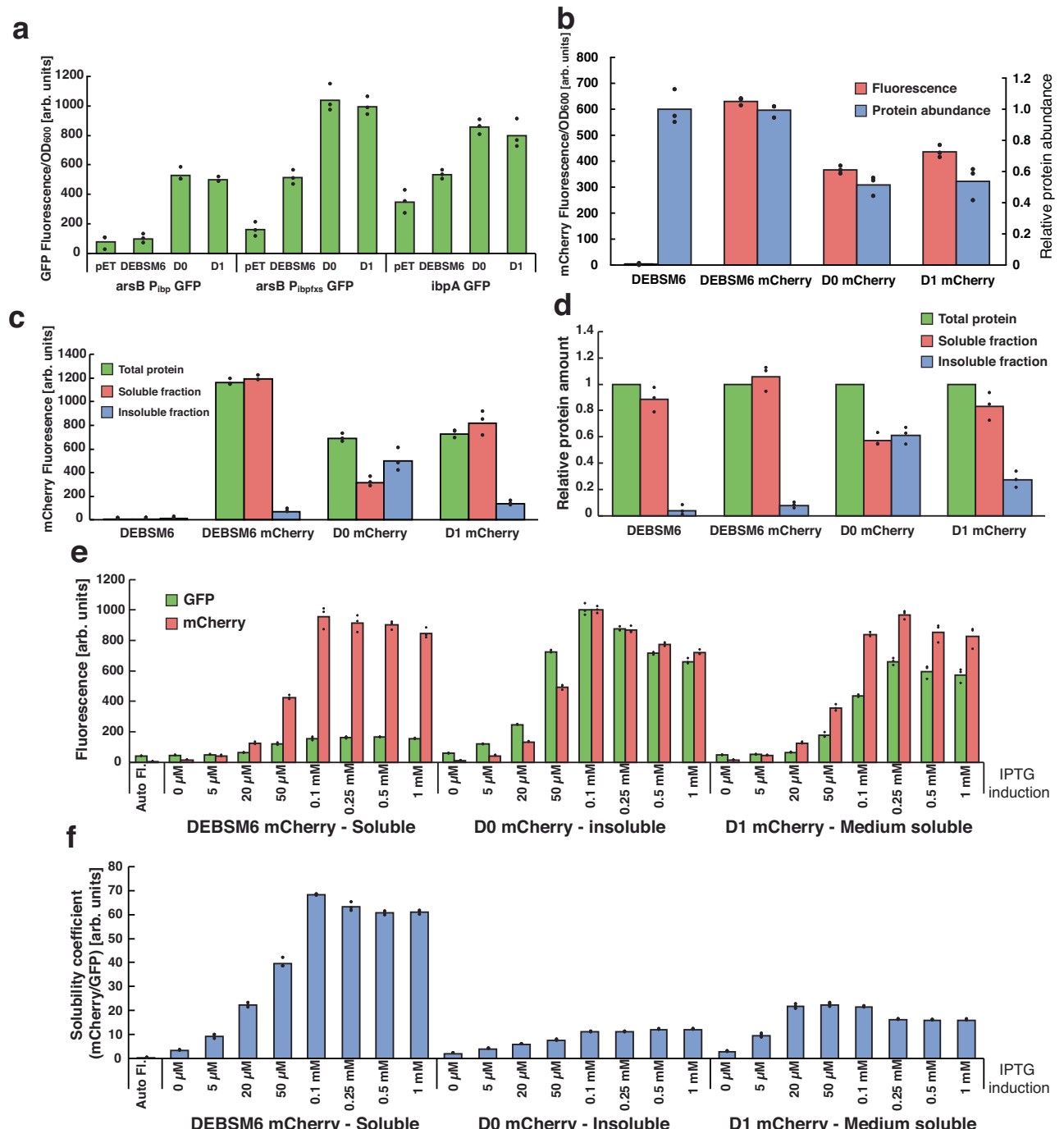

**Fig. 1 | Activity of misfolded protein biosensor and mCherry fusion proteins.**
**a** GFP fluorescence of *E. coli* biosensor strains expressing PKSs with variable solubilities. pET = pET28a (empty vector control). **b** Fluorescence of mCherry tagged PKSs (left y-axis) and SDS-PAGE quantified abundance of the proteins relative to DEBSM6 amount (right y-axis). **c** mCherry fluorescence and (**d**) SDS-PAGE quantification of PKS proteins in different protein fractions of lysed cells, relative to "Total protein" for each replicate. Cells were induced with 250 μM IPTG in (**a–d**). **e** Fluorescence of Δ*arsB*::P_ibp GFP strain expressing mCherry tagged PKSs. **f** Same results as 1e with a simplified "solubility coefficient": the ratio of the expressed protein (mCherry fluorescence) over activation of insolubility biosensor (GFP fluorescence). Data is presented as mean values of three biological replicates, dots are individual data points. In (**a–c**) fluorescence was measured using a microplate spectrophotometer, whereas in (**e**) fluorescence was measured using flow cytometry. Auto Fl = auto fluorescence, arb. units = arbitrary units. Source data are provided as a Source Data file.

simplify the presentation of solubility data, we define a solubility coefficient as mCherry fluorescence divided by GFP fluorescence. This coefficient measures how well a PKS is expressed relative to how much it misfolds and activates the biosensor (Fig. 1f). Thus, a PKS variant with high expression but low biosensor activation will have a high solubility coefficient score. For each figure in this report where the solubility coefficient is used, a corresponding figure can be found in the

supplementary information, which presents GFP and mCherry values separately (Supplementary Fig. 1-4).

### Constructing a randomized linker junction library
To create a randomized junction library of DEBSM6 engineered with EpoM4 AT, we first investigated the linker regions. EpoAT uses malonyl-CoA or methylmalonyl-CoA[31], unlike DEBSM6 AT, which uses

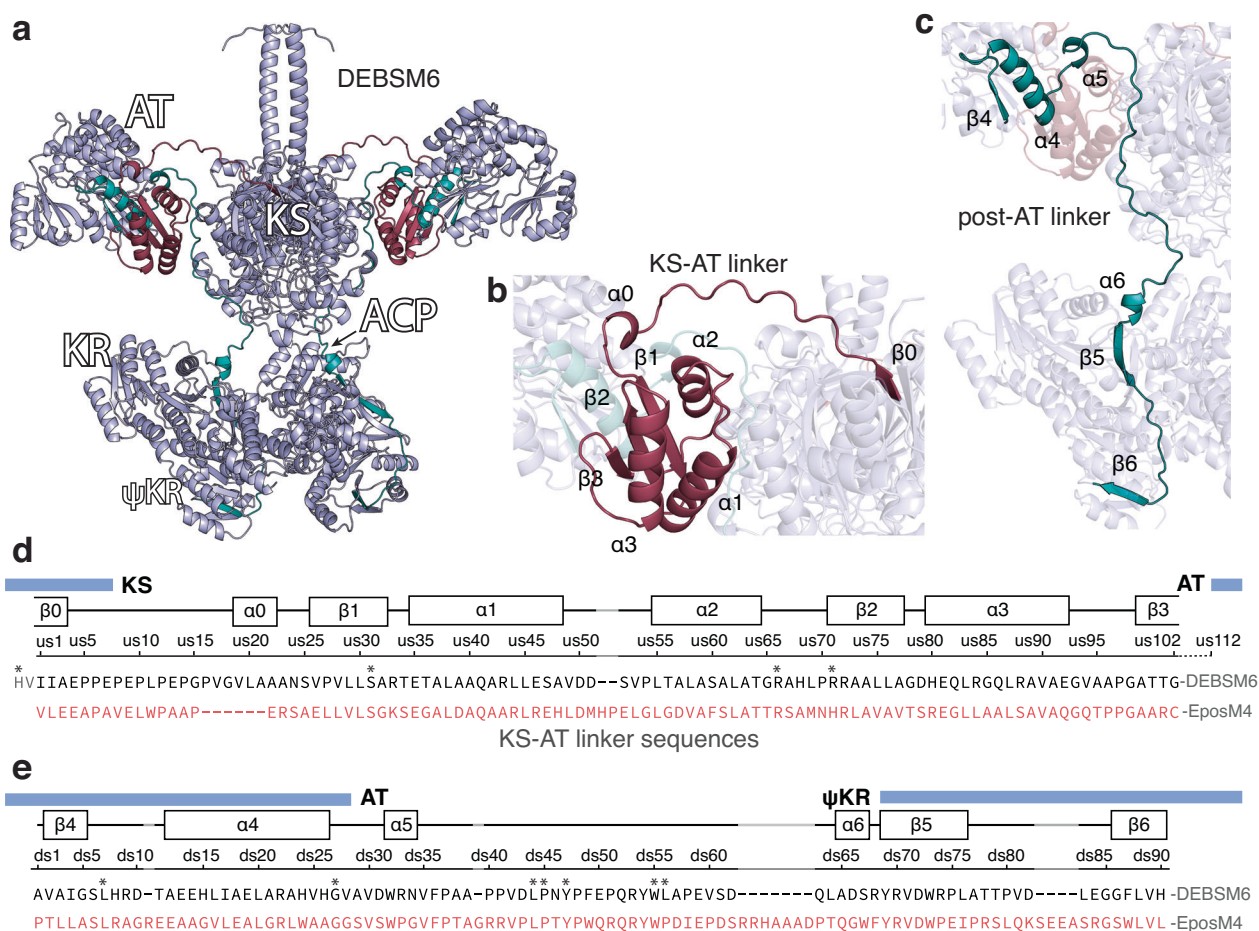

**Fig. 2 | Predicted structure and sequence of DEBSM6 and KS-AT and post-AT linker. a** AlphaFold structure prediction of homodimeric DEBSM6 without TE, (**b**) a highlighted structure of the KS-AT linker in dark red, and (**c**) the post-AT linker in teal. The regions highlighted are the sequences selected for the junction library. **d** Alignment of the region selected for the junction library of DEBSM6 and EpoM4 KS-AT and (**e**) post-AT linker with secondary structure elements predicted by AlphaFold. Domain boundaries marked with blue boxes were predicted using an online tool[49], except the start of ψKR which was placed at the beginning of β5. Each junction position in KS-AT linker is sequentially named us1-102 and post-AT junctions called ds1-90. Highly conserved residues are marked with asterisks. DEBSM6 KS-AT linker sequence starts with <u>HV</u> to denote where the conserved GTNAH sequence is positioned. Gaps in alignments are marked in grey. For reference, the position of the first amino acid in the KS-AT sequence is I1908 for DEBSM6 and V1948 for EpoM4 and in the post-AT linker, the first position is A2301 and P2345, respectively.

only methylmalonyl-CoA[8]. To get a structural understanding of the linker regions around the DEBSM6 AT, we used AlphaFold[32] to generate a homodimeric structure model (Fig. 2a, Supplementary Fig. 5). The AlphaFold results from the full DEBSM6 module could not predict the position and structure of the TE domain. Therefore, a model that excluded the TE domain was used. The KS-AT domain of the generated DEBSM6 model showed a high degree of similarity with the experimentally solved structures of KS-AT didomains from DEBSM3 and DEBSM5 (56% and 57% sequence identity with DEBSM6 KS-AT)[33,34] as well as the PKS module structures recently reported[16,17]. Promisingly, the generated model contains symmetrical dimeric KS-AT and asymmetrical dimeric KR-ACP, consistent with what was only recently reported from full module structures[16,17]. The predicted model's KS-AT linker consists of a disordered region starting from the KS, which then forms three alpha helices surrounding three beta strands (Fig. 2b, d). The highly ordered structure of the KS-AT linker suggested that it plays a more significant functional role than merely connecting two domains. In fact, interactions between the KS-AT linker structure and the ACP have been observed during chain elongation[16,17,35,36]. The AlphaFold model prediction of the post-AT linker, on the other hand, wraps itself around the residues of the KS-AT linker and continues

along the KS domain, interacting with several residues until it reaches the structural subdomain ψKR[37] (Fig. 2c, e).

Next, we developed an in vitro method for creating the randomized junction library of DEBSM6 engineered with the AT from EpoM4. Each variant of the library was designed to have the upstream and downstream junction randomly positioned somewhere in the KS-AT linker and post-AT linker respectively (Fig. 3a). To make the library, we used pooled oligonucleotides of up to 350 base pairs in length. The oligo pool library was designed by first selecting regions in the KS-AT linker and in the post-AT linker. Each amino acid in these regions corresponded to a specific placement of the DEBSM6-EpoM4 junction and were sequentially named us(upstream)1-102 (Fig. 2d) and ds(downstream)1-90 (Fig. 2e). Due to length limitations on the oligos, the entire KS-AT linker region could not be selected and the ten positions closest to the AT domain were excluded (Fig. 2b, us103-us112). In contrast, the post-AT linker was short enough that sequences outside of the linker region (residues inside the AT and ψKR domain) were also included in the library (Fig. 2c). Next, we designed one oligo per unique junction position from the selected regions by aligning the DEBSM6 sequence with EpoM4. Although the selected KS-AT linker region was 102 amino acids long, only 72 oligos had unique sequences

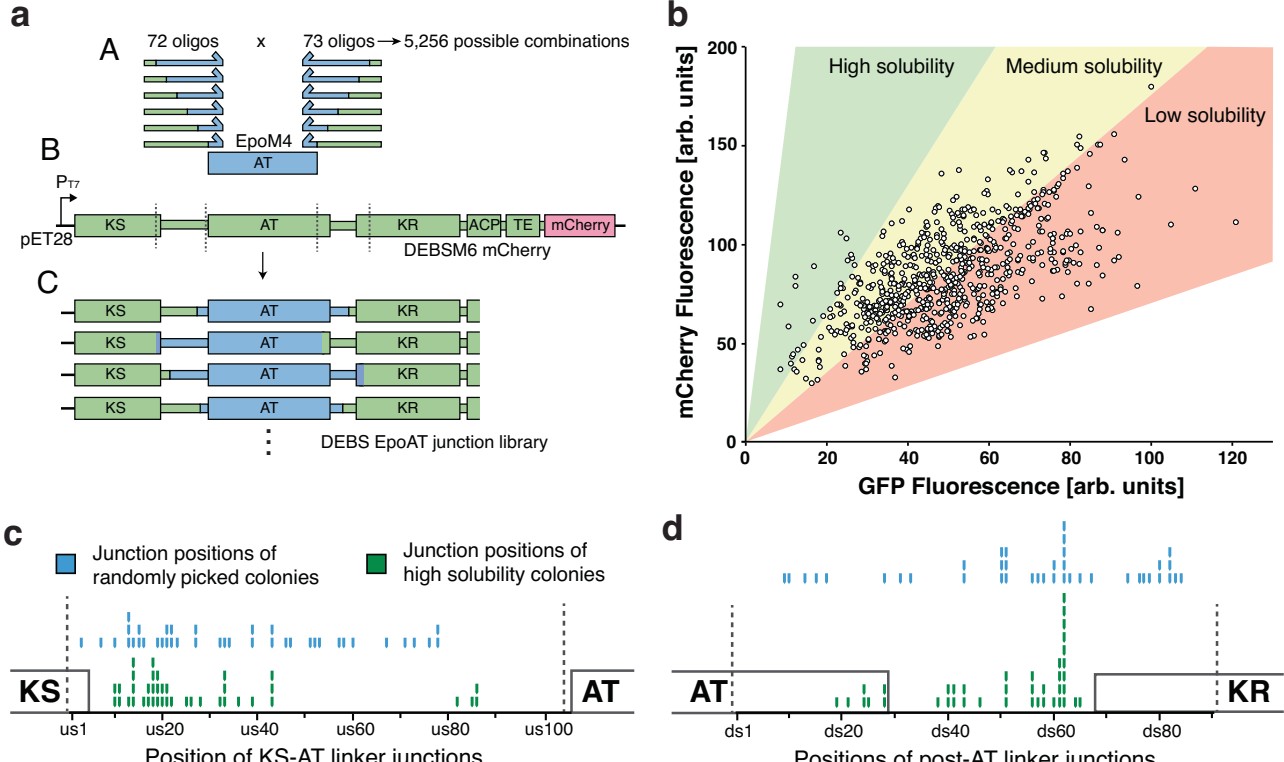

**Fig. 3 | Creation and measurement of randomized junction library. a** Strategy for creating junction library: (A) Pooled libraries of oligos, that each has the junction (where green DEBSM6 sequence and blue EpoAT sequence meet) positioned at a different place in the linker region, amplifies EpoM4 AT with PCR and are inserted into (B) DEBSM6 mCherry with the native AT excised. The resulting library (C) consists of a randomized upstream and downstream junction, with 5256 possible combinations. **b** Fluorescence measurement of junction library in Δ*arsB*::P$_{ibp}$ GFP strain using flow cytometry. Each dot represents one measured colony. Colored areas are estimations where differently soluble variants would appear. **c,d** Comparing junction positions between randomly picked colonies (blue sticks) with high solubility colonies (green sticks) in the KS-AT linker (**c**) and in the post-AT linker (**d**). Dotted line denotes selected linker region. Certain junction positions (e.g. ds62) were overrepresented due to how the library was designed: An alignment of DEBSM6-EpoM4 was used to decide junctions. Where there are gaps in the alignment (see position ds62 in Fig. 2e), the same start position of DEBSM6 was used for several end positions of EpoM4. This led to 8 unique variants all sharing ds62 as a junction. Source data are provided as a Source Data file. Arb. units = arbitrary units.

due to the presence of conserved regions where DEBSM6 and EpoM4 have stretches of identical amino acid sequences. In total, 72 unique oligos were created for the KS-AT linker and 73 for the post-AT linker, thereby resulting in 5256 possible combinations when randomly combining an upstream and downstream junction.

After designing and synthesizing the oligo pool library, we amplified EpoAT using PCR with the oligos with KS-AT junctions as "forward" primers and post-AT oligos as "reverse" primers (as shown in Fig. 3a). The resulting fragment mixture was then cloned into the AT position of DEBSM6 mCherry via Gibson assembly. Many of the resulting colonies of the library carried small deletions in the linker regions, possibly due to synthesis errors caused by the length of the oligos. Roughly 40% of colonies were visibly red indicating the presence and in-frame expression of mCherry.

### Biosensor-guided screening of soluble PKSs
Around 800 colonies with a visible red color were picked and grown in 96-well plates, and fluorescence was measured using flow cytometry (Fig. 3b). From this screening, we selected forty colonies with high solubility, and their corresponding plasmids were purified and sequenced to determine what junction positions were enriched in the high solubility set (Supplementary Table 1). For comparison, we also sequenced a reference set of 40 random colonies with undetermined solubility. Interestingly, while the reference set showed a mostly even distribution of junctions, except for positions us80-us102 which appeared less frequently than expected, the high solubility set exhibited a clear bias against junctions at certain positions (Fig. 3c, d).

Specifically, we found that no high soluble variant had their junctions in positions us46-81, ds1-18 and ds65-90, indicating that junctions in those positions negatively affected solubility. The gap between us46-81 was predicted by AlphaFold to contain the alpha helix 2 (α2) and the beta strand 2 (β2), both of which are deeply embedded within the KS-AT linker structure (Fig. 2b). Ds1-18 and ds65-90 place the junction inside the AT structure (Fig. 2d) or inside the ψKR domain (Fig. 2e). Disturbance of either of these domains would likely cause the PKS to lose stability and become insoluble.

The 40 library colonies that showed high solubility were remeasured in triplicate (Fig. 4a) and a subgroup of five were selected to assess protein solubility and enzyme productivity: o4 (us85/ds21), o8 (us13/ds43), o15 (us21/ds64), o17 (us28/ds62) and o33 (us17/ds25) (Supplementary Fig. 6). Among these five variants, o33 was found to be the most soluble, with a solubility coefficient close to that of wild type DEBSM6. To investigate the solubility of these proteins in the absence of mCherry, plasmids were constructed expressing the variants without the C-terminal fusion tag, and protein amounts in the soluble and insoluble fractions were quantified by SDS-PAGE (Fig. 4b). The insoluble fractions of the five selected variants (o4 10.4%, o8 8.6%, o15 13.5%, o17 13.1%, o33 13.7%) were all lower than the references (D0 63.6%, D1 19.4%), confirming that the selected variants have improved solubility.

### In vitro productivity assay of high solubility variants
The five highly soluble variants, DEBSM6 and D1 (us1/ds44) were purified using nickel affinity resin (Supplementary Fig. 7a). In vitro

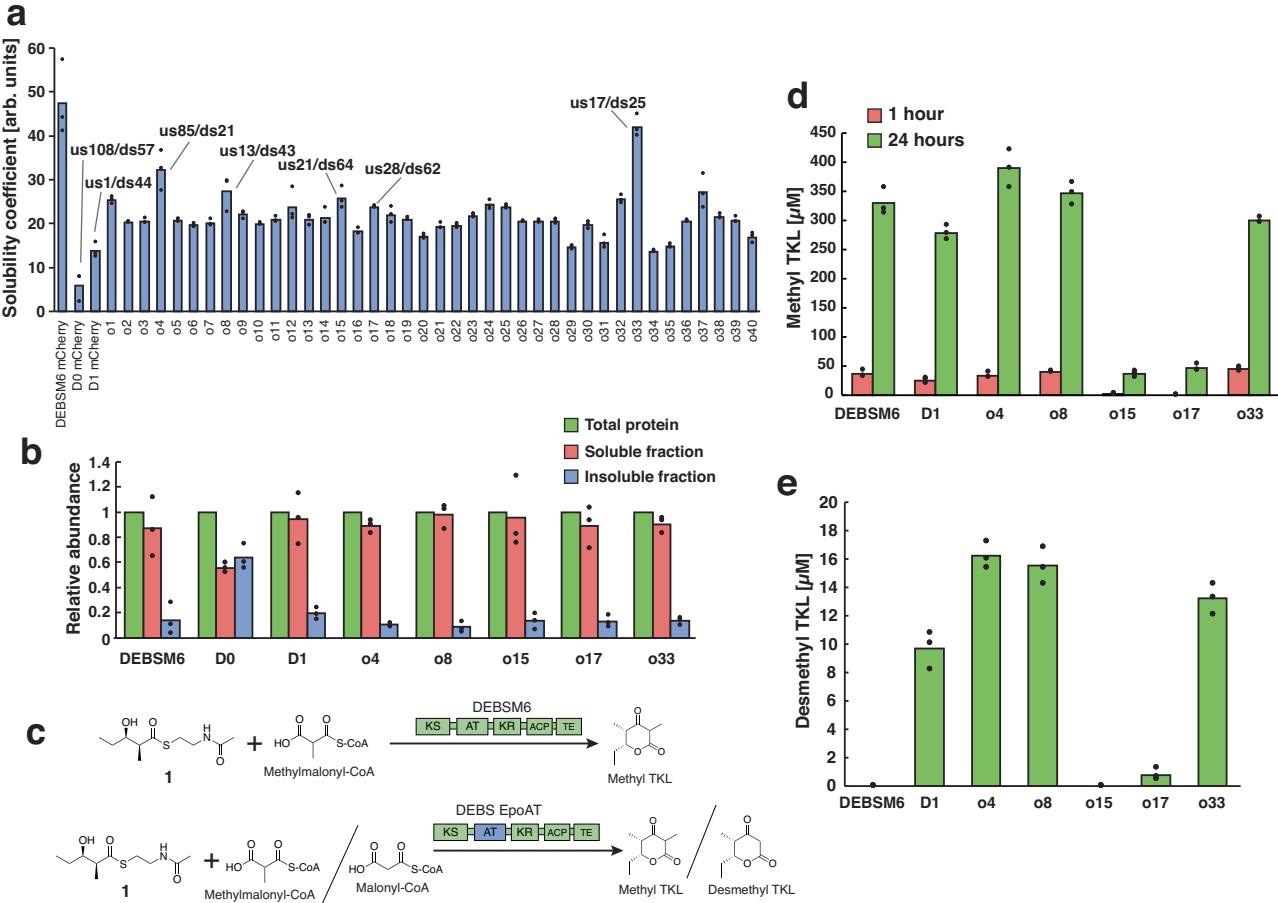

**Fig. 4 | Solubility and in vitro productivity of high solubility library variants.**
**a** Solubility measurement using $\Delta arsB$::$P_{ibp}$ GFP biosensor strain of 40 high solubility colonies, a subset of which have their junction positions marked and were selected for (**b**) SDS-PAGE quantification of PKS abundance in different protein fractions. (**c**) In vitro reaction scheme with the synthetic starter **1** and methyl- or malonyl-CoA as substrate. DESBM6 AT natively accepts methylmalonyl-CoA while EpoM4 AT can accept both methyl- and malonyl-CoA. In vitro production of methyl TKL after 1 and 24 hours (**d**) and desmethyl TKL after 24 hours (**e**). DEBSM6 is the parental PKS, D1 is included as a reference to what is currently known as the optimal junctions for domain exchange. D0 was excluded due to it already been shown to be inactive[8]. All strains were induced with 250 μM IPTG. Data is presented as mean values of three biological replicates, dots are individual data points. Arb. units = arbitrary units. Source data are provided as a Source Data file.

enzymatic productivity was measured using a synthetic starter substrate (**1**) and either malonyl-CoA or methylmalonyl-CoA as extension substrates, resulting in a desmethyl or methyl triketide lactone (TKL), respectively (Fig. 4c). Reactions were sampled at 1 and 24 hours for methyl TKL and at 24 hours for desmethyl TKL, and product titers were quantified using authentic standards (Supplementary Fig. 8). Three variants, o4, o8 and o33, and D1 produced methyl TKL in titers similar to the wild-type DEBSM6 indicating that protein structures of o4, o8 and o33 are not destabilized even with a heterologous AT domain (Fig. 4d). The junction positions in the KS-AT linker were at the beginning of the linker (β0-α0) for o8 (us13) and o33 (us17), or in the middle of α3 for o4 (us85). These junctions either included the entire KS-AT linker from EpoM4 or retained the counterpart of the parental PKS. The downstream junctions in the post-AT linker were positioned at the end of the AT domain for o4 and o33 (ds21 or ds25) or just before the residues interacting with the KS domain for o8 (ds43). Both o15 and o17 had high solubility but showed significantly lower production. These barely active variants had the downstream junction at ds62 and ds64, respectively, meaning residues within the post-AT linker that interact with the KS (ds44-56 in AlphaFold prediction) had the heterologous EpoM4 AT sequence. This part of the linker is known to tightly interact with the KS in DEBS[33,34] and in other PKSs[16,38], and is critical for KS condensation reaction[39], indicating that the heterologous

linker sequence in the o15/o17 variants is unable to complement that function[40].

When malonyl-CoA was used as a substrate for the in vitro reaction, no production was observed for DEBSM6, as expected, since the native AT cannot accept malonyl-CoA[8] (Fig. 4e). In contrast, o4, o8 and o33 produced more desmethyl TKL than the less soluble reference D1, albeit at a ~23 times lower amount compared with methyl TKL production. This is consistent with previous results that have shown DEBSM6 to be less accepting of malonyl-CoA compared with the native methylmalonyl-CoA substrate. This is thought to be due to gatekeeping by the rest of the PKS domains, e.g., the KS[8].

## Investigating the effect of junction positions in KS-AT linker on solubility

Next, we selected eight junction positions that were evenly spaced out in the KS-AT linker and three in the post-AT linker and constructed all 24 combinations. Upon measuring fluorescence, results show that positions us85, us28 and us17 always led to the highest relative solubility, no matter if the downstream junction was in a position that contributed to high (ds25), medium (ds62) or low solubility (ds81) (Fig. 5a). These results indicate that junctions influence protein stability independently of each other. Therefore, it is possible to assess an upstream junction's general influence on stability, independent of the paired downstream junction.

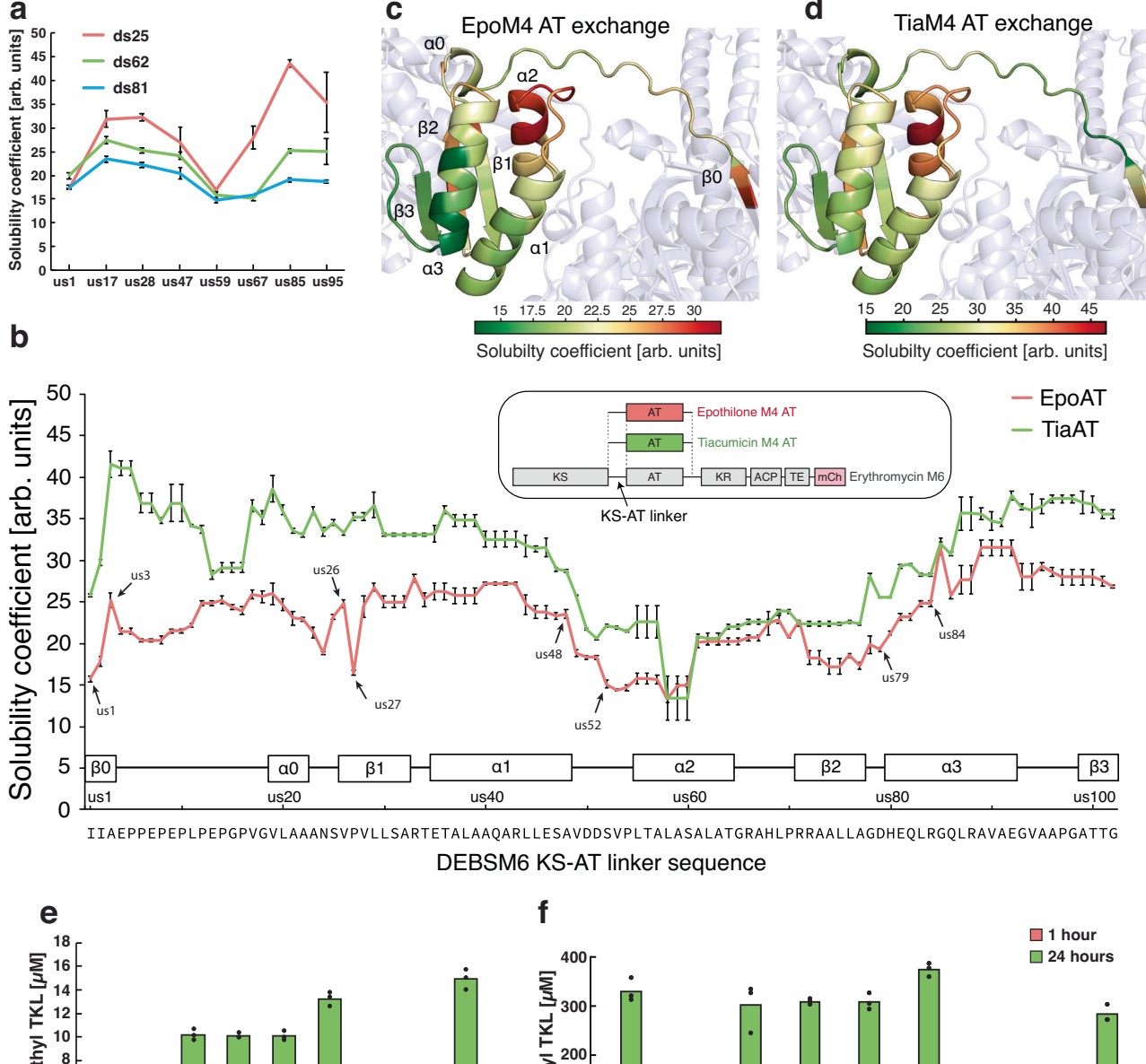

**Fig. 5 | Effect of KS-AT linker junction positions on solubility and product formation. a** Testing synergy between KS-AT and post-AT linker junctions. The solubility coefficient was measured using the $\Delta arsB$::$P_{ibp}$ GFP strain harboring DEBSM6 mCherry engineered with EpoM4 AT with different junction combinations. The color of the lines indicates the position of the post-AT junction, while the x-axis labels indicate the position of the KS-AT junction. **b** KS-AT junction effect on solubility coefficient measured using $\Delta arsB$::$P_{ibp}$ GFP strain harboring DEBSM6 mCherry engineered with EpoM4 AT or TiaM4 AT with downstream junction at ds25. Sample pairs tested for in vitro production in (**e**) and (**f**) are marked with an arrow. Data points from neighboring junction positions that give identical

polypeptide sequence due to homology between the parental PKS sequence and exchanged AT sequences are repeated. For reference, the DEBSM6 mCherry solubility coefficient was 71.1 ± 0.5, and its expression was induced with 250 μM IPTG. **c, d** Solubility data from (**b**) visualized on predicted protein structure of DEBSM6 engineered with EpoM4 AT (**c**) or TiaM4 AT (**d**). **e, f** In vitro production of des-methyl TKL (**e**) and methyl TKL (**f**) of variant pairs of DEBSM6 with EpoM4 AT. Reactions were sampled at 1 hour (red bars) and 24 hours (green bars). Data is presented as mean values of three biological replicates, dots are individual data points. Arb. units = arbitrary units. Source data are provided as a Source Data file.

To do just that, we tested every amino acid position in DEBSM6 KS-AT linker for their general effect on solubility. We selected ds25 from the most soluble o33 variant as downstream junction and constructed 72 AT exchanged PKSs, each with a different KS-AT junction from the same region as the oligo pool library (positions us1-us102). The reason why 72 constructs could cover all unique junctions in a 102-amino acid long sequence is due to homology between DEBSM6 and

EpoM4 linker sequences. Junction positions us97-us100, for example, result in the same APGA sequence (Fig. 2d). Therefore, only one construct was made per non-unique junction. The AT from EpoM4 was again used for the domain exchange. The 72 engineered PKSs were transformed into $\Delta arsB$::$P_{ibp}$ GFP, and solubility was assessed using the solubility coefficient. Surprisingly, most junction positions in the KS-AT linker showed relatively high solubilities (Fig. 5b, c). Certain regions

in the linker, such as the α2 and β2 structure showed large reduction in solubility, which is strikingly consistent with our results from the randomized library (Fig. 3c). Junctions often had a similar solubility as their neighbors with a notable exception of one proline residue in β1 (us27), which had a significant drop in solubility coefficient, while its neighboring positions did not.

To further investigate the solubility-activity relationship, four variant pairs were selected with nearby junctions but with differences in solubility: us1/ds25, us3/ds25, us26/ds25, us27/ds25, us48/ds25, us52/ds25, us79/ds25, and us84/ds25. These PKSs were purified without the mCherry fusion tag, and all proteins were isolated with high purity except us52/ds25 and to a lesser extent us64/ds25 (Supplementary Fig. 7b). In vitro analysis of desmethyl and the methyl TKL production showed that the less soluble mutant pair made less product in all examples but one, and these results were consistent for both TKL products (Fig. 5e, f). The one exception was us27 that had the junction in the aforementioned β1 proline. Because all tested variants have the same post-AT linker junction (ds25), the cause of the structural destabilization and low production can be attributed to an unfavorable junction in the KS-AT linker. These results demonstrate that small changes in placement of junctions can have a major effect on the activity of the enzyme. The us48 and us52 variants only differed by 4 amino acids but us48 had wild type level of production while us52 was inactive. The results also indicate that a junction giving a more soluble PKS has a better chance to be active than a PKS engineered with a destabilizing junction, although this is not a strict correlation as us26 and us27 had identical production. This agrees with the idea that protein function follows protein structure.

To see how generalizable these results are, we repeated the experiment with a phylogenetically distant AT, the ethylmalonyl-CoA–specific AT from the tiacumicin PKS module 4 (TiaM4)[41]. In this case, 69 unique constructs were made to cover the same linker region. Results from the fluorescence measurement show that the solubility patterns from TiaAT exchanged DEBSM6 agree remarkably well with the corresponding EpoAT results (Fig. 5b, d). Again, upstream junctions within α2 and β2 appear to destabilize the protein while areas us3-us40 and us90-us102 give in general high solubility. These results indicate that the junction position in DEBSM6 has a bigger impact on protein stability than the sequence of the AT domain being inserted.

## Discussion

Type I modular PKSs are multi-domain enzymes that produce various polyketide structures by combining several catalytic domains in a specific order in an understandable way. Recent studies suggest that tens of thousands of type I modular PKSs and hybrid PKS-nonribosomal peptide synthetases are encoded in genomic sequences, indicating that nature utilizes these enzymes to produce diverse molecules on demand. Since the early 1990's, significant efforts have been made to rationally engineer type I modular PKSs by domain exchange. Although around a hundred natural polyketide analogs have been generated by rational protein engineering, unfortunately the resulting engineered PKSs generally have significantly reduced kinetics compared with their wild-type counterparts, producing low titers of desired molecules[6,7]. Several reasons contribute to why domain exchange causes PKSs to lose activity, such as poor specificity towards the altered substrate by the other domains in the module and the loss of protein structure due to incorrect domain boundaries selected[8,42]. One significant barrier that slows PKS engineering progress is the lack of a general method to screen active versus inactive PKS variants in an engineered PKS library: products from type I modular PKSs usually do not absorb light or fluoresce. Although a recently developed polyketide biosensor holds promise for that, it is not practical to design a biosensor for every target molecule[43].

In the present study, we developed and characterized a biosensor that can discriminate between soluble and insoluble PKS variants. This biosensor can detect if the selected domain boundaries cause disruptions to the structure of the enzyme. In the first screening from ~800 AT-exchanged PKSs where junctions in the KS-AT linker and the post AT liner were randomized, we selected 40 variants with high solubilities (Fig. 4a). In vitro activity assay demonstrated that 60% of the highly soluble AT-exchanged PKSs showed a wild-type level of catalytic activity. As expected, these variants also produced a polyketide product from a nonnative substrate. The most soluble variant from the initial screening was o33 where the junctions in the KS-AT linker and the post AT linker were us17 and ds25, respectively. The us17 junction in the KS-AT linker is located between β0 to α0, an unstructured region in the linker. We used AlphaFold2 to compare 15 additional PKSs, and this region consists of a random coil in all models except for the bafilomycin PKS module 5 where there is an insertion of ~25 amino acid residues (Supplementary Table 2, Supplementary Fig. 6). These structural models also show that the triple alpha helix motif (α1-α3) in the KS-AT linker is conserved except for the aculeximycin PKS M7 and the rifamycin PKS M1. Both structures lack α3 between β2 and β3, which could be seen as gaps in the sequence alignment. The triple beta sheet motif is also conserved in all analyzed PKSs except for the borrelidin PKS M1 where β3 was missing. For the post-AT linker, all predicted modules exit the AT domain with α4, after which the linkers wrap around KS-AT linkers and continue alongside the KS domains where LPTY(A/P)FQ(H/R)xRYWL motif binds to the KS surface after which the models diverge depending on what domain that follows. These models appear to explain why ds21-25 (o4 and o33) and ds43-44 (o8 and D1) variants retain activities, but ds62-64 variants (o15 and o17) lose activities.

In our detailed investigation of junctions in the DEBS KS-AT linker, we observed that most variants are highly soluble except for the ones that have the upstream junctions between α2 and β2 structure regions, which are less soluble and less active (us53 and us79). In contrast to our previous suggestion for the optimal domain junction (=us1/ds44), which was determined by in vitro kinetic analysis of a series of AT-exchanged PKSs[8], the current findings indicate the position of domain boundaries in KS-AT linker can be somewhat flexible over a range of positions. We also observed a striking resemblance of the solubility pattern from EpoAT and TiaAT (Fig. 5b), implying that the position of the junction in DEBSM6 was more impactful than differences in the AT sequences that were inserted. Additionally, domain exchange with TiaAT gave, on average, a higher solubility coefficient than when EpoAT was used, indicating that the AT domains themselves influence solubility regardless of which domain boundaries are used.

The use of this biosensor method also has some important limitations. One issue is that high solubility does not guarantee activity, as many other factors play a role. This is illustrated well by the difference in production of methyl TKL and desmethyl TKL for the AT replaced DEBSM6 (Fig. 4d, e). Despite using domain boundaries that minimally affected protein stability, the production of the desmethyl product was still ~5% that of the non-native product. Improving domain junctions is unlikely to increase the production of desmethyl TKL further. Instead, increasing the promiscuity of the other domains in the module through point mutations would likely be necessary. However, our methodology can at least remove structurally disrupted variants from a domain exchanged PKS library. These variants are relevant to remove, as in 3 out of 4 cases, we observed that less soluble variants made less product than nearly sequence identical high solubility variants (Fig. 5e, f).

Another limitation is that some PKSs might already activate the biosensor to its maximum extent. Domain exchanges into those PKSs may not result in noticeable changes in biosensor activation. Under such circumstances, truncating the PKS and expressing only a few domains could potentially provide a better dynamic range for the biosensor. Lastly, comparing solubility coefficient scores between two unrelated PKSs can lead to inaccurate results due to differences in

protein expression and base activation of biosensor. Normalizing by protein expression may not always resolve this issue, as a PKS with twice the expression may not necessarily generate a proportional increase in the biosensor signal. Despite these limitations, we believe the biosensor can be a useful tool for PKS engineering when applied to the appropriate task.

In summary, based on in vivo solubility data and in vitro activity data, as well as structure data predicted by AlphaFold2, we revise our previously suggested boundaries for AT domain exchange ( = us1/ds44). The above observations suggest that i) when the ds44 junction is selected us3-us17, not us1, should be selected as upstream junctions and ii) when the ds25 junction is selected us3-us17 or us84-us102 should be selected as upstream junctions. In many cases, attempting multiple junctions may not be feasible, and in such instances, we recommend using us3/ds44 (Supplementary Fig. 6). To locate these sites, the respective parental and donor PKS can be aligned with the primary sequence of DEBSM6. Secondary structure features of experimentally verified and AlphaFold2 models of KS-AT domains aligned well (Supplementary Fig. 7). This indicates that structure prediction or experimental data is not necessarily required to locate these secondary structural elements, but a sequence alignment should, in most cases, suffice. Our work suggests that this system could also be used with other PKS domains such as KS, KR, DH, and/or ER domains to fully optimize domain boundaries of domain exchanged PKSs.

## Methods

### Cloning and cell cultivation
The sequence of plasmids can be found in Supplementary Table 3. Each plasmid was constructed using some combination of restriction enzyme digestion and Gibson assembly[44]. Positive colonies were confirmed using Sanger sequencing. All cell cultivation was done in lysogeny broth (LB, Miller) supplemented with 50 ug/ml kanamycin sulfate (Teknova, USA), and plate selection was done on LB + kanamycin agar plates (Teknova).

### Protein fractionation and SDS-PAGE quantification
The *E. coli* strain BAP1[45] was used for expression of proteins. BAP1 is based on BL21 (DE3) and expresses a phosphopantetheinyl transferases that is necessary for purifying PKSs in their holo-form. Plasmids were transformed into BAP1 and selected on LB plates. Colonies were picked in triplicates and grown at 37 °C. The next day fresh LB was inoculated 1:100 by volume with overnight cultures and grown at 37 °C for 2 hours until $OD_{600}$ ~ 0.5. Cultures were put on ice for 30 min to stop growth, induced with 250 μM IPTG (Teknova), then grown overnight at 18 °C. The following day, $OD_{600}$ was measured, and the same amount of cells was taken for each culture equal to around 1.5 ml of overnight culture. Cells were centrifuged and the pellet was resuspended in 500 ul phosphate-buffered saline (PBS). Samples were sonicated with a Q125 sonicator (Qsonica, USA) at 30% amplitude for 5 seconds twice to lyse the cells and a small sample was saved as the "total protein fraction". Samples were centrifuged at max speed (~21,000 g) for 2 min and the supernatant was saved as the "soluble fraction". The pellet was resuspended in an equal volume PBS as was removed and saved as the "insoluble fraction". If proteins were fused with mCherry, fluorescence was determined by measuring at 587 excitation/617 emission with a 610 excitation cut off on a SpectraMax M2e (Molecular Devices, USA). For SDS-PAGE, fractionated samples were mixed 1:1 with Laemmli Sample Buffer (Bio-Rad, USA) supplemented with 100 mM DTT (VWR Scientific) and boiled at 95 C. Samples were loaded onto Mini-Protean TGX 8–16% 12-well precast gel (Bio-Rad) and run for 30 minutes at 200 volts. Gels were washed twice in boiling water and then stained with GelCode Blue Safe Protein Stain (Thermo Scientific, USA). Protein bands were quantified using the software VisionWorks (Analytik Jena, Germany) by first measuring the relationship between pixel intensity and loaded protein using a standard curve created using a series of 2-fold diluted samples. Relative protein content in each fraction was then calculated by dividing by the total protein fraction for each individual replicate.

### Integration of biosensor into the genome
The biosensor strains were engineered using the λ red recombinase protocol described in Datsenko and Wanner 2000[46]. For the ΔarsB::Pibp GFP and ΔarsB::Pibpfxs GFP strain, the promoter(s), *gfp*, a kanamycin cassette and an upstream and downstream homology region of 1000 base pairs were amplified using PCR and combined into a single fragment with Gibson assembly. For the ΔibpA::GFP strain, the same method was used except that the promoter sequence was already located on the upstream homology region (see Source Data file for DNA sequences). The combined DNA fragments were used to transform chemically competent BL21 (DE3) cells carrying pKD46 plasmid, induced with 0.1 % arabinose to express recombinase genes. After positive colonies were verified by colony PCR screening, the kanamycin cassette was removed by transforming in the FLP recombinase on the pCP20 plasmid. The plasmid was then cured away by growing the cells at 43 °C.

### Induction of biosensor + measurements
Purified plasmids were transformed into ΔarsB::Pibp GFP and plated on LB agar plates. For the oligo pool library, the casted agar with colonies was carefully lifted and IPTG was added to the bottom of the plates to a concentration of 50 μM assuming a content of 20 ml LB agar. The plates were incubated at room temperature for a day until colonies turned visibly red. Either red colonies for the oligo pool library or uninduced colonies for the other experiments were grown in 300 μl LB in 96-well deep plates as seed cultures. The next day, 1% was seeded into 300 μl LB and grown at 37 °C for 2 hours, put on ice for 30 min, induced with 250 μM IPTG (if not otherwise noted) and grown at 18 °C, 250 rpm overnight. If several 96-well plates were grown at the same time, cultures were started with a 1 hour delay so that they could be measured after an equally long induction. The following day after ~20 hours of induction, samples were measured using flow cytometry (BD Accuri C6). The flow cytometer collected 50000 events larger than 2000 FSC-H per sample and GFP and mCherry fluorescence was measured. At 250 μM IPTG induction, mCherry fluorescence of cells showed a bimodal distribution with more than half expressing mCherry and the rest not (Supplementary Fig. 11). Gating was used to calculate the average mCherry and GFP fluorescence of the mCherry containing cell subpopulation. The solubility coefficient was calculated as mCherry divided by GFP fluorescence.

### Design of oligo pools and construction of swap junction library
The oligo pool nucleotide sequences used to create the swap junction library can be divided into three parts: a shared 5' region that was used to enrich for full length sequences and had an overlap with the DEBSM6 plasmid, a variable region that is unique for each oligo and encoded the swap junction somewhere along its sequence, and a shared 3' region which anneals to the EpoM4 AT sequence. For the variable region, DEBSM6 and EpoM4 were aligned using Clustal Omega and homology sequences in the KS-AT and post-AT linkers were selected corresponding to roughly 300 base pairs in length each[47]. To make sure each oligo contained a unique junction, only one was selected in regions of homology between EpoM4 AT and DEBSM6 AT where adjacent junctions resulted in identical protein sequences. In total, the "forward" KS-AT oligo pool library contained 72 oligos and the "reverse" post-AT library contained 74 oligos. The oligo pool libraries were synthesized by Integrated DNA Technology (IDT, USA). For a complete sequence list, see Supplementary Data 1.

For the library construction, the EpoM4 AT sequence was amplified using the two oligo pool libraries using standard PCR conditions. The correct size band was gel purified and further enriched by using it

as a PCR template with primers binding to the shared 5' sequence. The resulting product was cloned into DEBSM6 mCherry digested with KpnI and BamHI. The library was then transformed into XL1-blue, purified and then transformed into the ΔarsB::P_ibp GFP strain.

### Protein structure prediction using AlphaFold

Structure prediction of the homodimeric DEBSM6 was performed utilizing an adapted version of the AlphaFold program[32]. Modifications to enable homooligomer modeling were adapted from the ColabFold project[48]. Additional code to run AlphaFold program can be downloaded from https://github.com/Alberto024/alphafold. Final DEBSM6 structure prediction was performed without the thioesterase, as AlphaFold was unable to generate a structure with the correct placement of the thioesterase, for a total of 3360 amino acids. The best structure achieved a plddt of 87.9 using 2626 seconds for feature generation, 200 seconds for feature processing, and 883 seconds for model compilation and prediction.

For the additional PKSs modeled, we selected 15 modules with a high degree of domain diversity. These included modules with varying amounts of reducing domains, AT substrate specificities and with and without KR dimerization elements. To get models with high confidence scores, the sequence of the KS, AT and subsequent domain was input into AlphaFold. See Supplementary Table 2 for a list of PKSs, the Source Data file for PDB files.

### Protein purification

All purified proteins had the C-terminally fused mCherry tag clonally removed and the resulting plasmid transformed into BL21 (DE3). Growth and induction were done the same way we previously described except cultures were grown in 1 L cultures. After growing overnight at 18 °C, cultures were harvested by centrifuging at 5000 g for 10 min and pellets were frozen and stored at -80 °C until future use. Proteins were purified as described before[8]. Thawed cell pellets were resuspended in lysis buffer (50 mM phosphate buffer pH 7.6, 50 mM NaCl, 10 mM imidazole), lysed by sonication and centrifuged at 8000 g for 15 min three times to remove cell debris. The supernatant was mixed with 2 mL HisPur Ni-NTA resin (ThermoFisher Scientific) and incubated at 4 °C for 1 hour. Next, the protein-resin mixture was washed three times with lysis buffer and eluted with elution buffer (150 mM phosphate buffer pH 7.6, 50 mM NaCl, 150 mM imidazole). The eluted proteins were then injected into an Äkta Explorer (Cytiva, USA) and captured on the anion exchange column HiTrap Q HP 5 mL (Cytiva), washed with 5 column volumes of wash buffer (50 mM phosphate buffer pH 7.6, 8% glycerol) and gradually eluted with anion elution buffer (50 mM phosphate buffer pH 7.6, 500 mM NaCl, 8% glycerol). Collected fractions were concentrated using Pierce Protein Concentrator 100 K (ThermoFisher Scientific), and then aliquoted, frozen and stored at -80C.

### In vitro assay

The in vitro reaction was based on a previously described protocol[8]. The following compounds were added to the reaction: 400 μM methylmalonyl-CoA tetralithium salt or malonyl-CoA tetralithium salt (both Milipore-Sigma, USA), 2 mM (2 S,3 R)-3-hydroxy-2-methylpentanoyl-S-N-acetylcysteamine thioester (1, synthesized as in Yuzawa et al.[8], see Supplementary Fig. 12 for synthesis scheme), 2 mM DTT, 100 mM phosphate buffer pH 7.2 and 1 μM PKS enzyme. To simplify the analysis, NADPH was omitted so that only the nonreduced product was made instead of a mixture of reduced and nonreduced product. Reactions were incubated at room temperature and samples were taken at 1 hour for the methylmalonyl-CoA reaction and 24 hours for both methyl- and malonyl-CoA samples. Samples were only taken at 24 hours for the reaction with malonyl-CoA due to too low product formation at 1 hour. In vitro products were extracted by adding two volumes of ethyl acetate and collecting the supernatant twice. After

evaporating the ethyl acetate, products were resuspended in 50% methanol and filtered through 3 K Amicon centrifugational filters (Merck, Germany).

LC separation of TKLs was conducted at room temperature with a Kinetex XB-C18 column (100 mm length, 3 mm internal diameter, 2.6 μm particle size; Phenomenex, USA) using a 1260 Infinity II LC System (Agilent Technologies). The mobile phase was 0.1% formic acid in water (solvent A) and 0.1% formic acid in methanol (solvent B). Products were separated at a flow rate of 0.42 mL/min using the following gradients: 20% to 72.1% B in 6.5 min, 72.1% to 95% B in 1.3 min and held for 1 min. Then, at a flow rate of 0.65 mL/min, 95% to 20% B in 0.2 min and held for 1.2 min. The LC system was coupled to an Agilent InfinityLab LC/MSD iQ single quadruple mass spectrometer (Agilent Technologies) and ESI was conducted in the negative-ion mode. SIM mode was run at mass 155 and 169 for desmethyl and methyl TKL respectively and products were quantified by comparing peak areas with authentic standards synthesized by Acme Bioscience.

### Statistics and reproducibility

No statistical method was used to predetermine sample size. No data were excluded from the analyses. The experiments were not randomized. The Investigators were not blinded to allocation during experiments and outcome assessment.

### Reporting summary

Further information on research design is available in the Nature Portfolio Reporting Summary linked to this article.

## Data availability

A list of all plasmids used in this paper can be found in Supplementary Table 3 and plasmid sequences and strain request is available at the Joint BioEnergy Institute's public Inventory of Composable Elements (https://public-registry.jbei.org). Source data is available as a Source Data file for all figures. Source data also contains pdb structures of AlphaFold models, DNA sequences of biosensors and raw data files from a subset of LC-MS samples. The complete data set from Fig. 5b with solubility values and protein sequences are available as Supplementary Data 2. Source data are provided with this paper.

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

## Acknowledgements

This work was performed as part of the US Department of Energy (DOE) Joint BioEnergy Institute (https://www.jbei.org) supported by the DOE, Office of Science, Office of Biological and Environmental Research, under contract DEAC02-05CH11231 between the DOE and Lawrence Berkeley National Laboratory and was additionally supported by

National Science Foundation grant 2036849 and by a DOE Office of Science Distinguished Scientist Award to J.D.K. E.E. was supported by Formas Mobility Grant Nr. 2017-00335. A.A.N. was supported by a National Science Foundation Graduate Research Fellowship, fellow ID 2018253421. S.Y. was supported by research funds from Yamagata Prefectural Government and Tsuruoka City, Japan.

The authors acknowledge the support of Alex Hexemer, Hari Krishnan and Peter Zwart via Center for Advanced Mathematics for Energy Research Applications (CAMERA), which is jointly funded by the Advanced Scientific Computing Research (ASCR) and Basic Energy Sciences (BES) programs in the Department of Energy Office of Science, under Contract No. DE-AC02-05CH11231 and via the Artificial Intelligence and Machine Learning at DOE Scientific User Facilities program under Award Number 107514 MLExchange.

## Author contributions

E.E. performed most experiments. E.E. and M.S. analyzed in vitro production. A.N. performed structure modeling. M.S, S.K and L.K. performed experiments during manuscript revisions. E.E., L.K., S.Y., J.D.K., Q.D., were responsible for experimental design. All authors contributed to the preparation of the manuscript.

## Competing interests

J.D.K. has a financial interest in Amyris, Lygos, Demetrix, Maple Bio, Napigen, Apertor Pharma, Ansa Biotechnologies, Berkeley Yeast, and Zero Acre Farms. L.K. has a financial interest in Lygos. All other authors declare no competing interests.
