## [Peer Review File · Nature Communications]

REVIEWER COMMENTS

Reviewer #1 (Remarks to the Author):

Review of Biosensor Guided Polyketide Synthase Engineering for Optimization of Domain Exchange Boundaries, by Englund, et al.

This paper reports the use of solubility-guided screening to identify improved sites for swapping of acyl transferase (AT) domains into a model modular polyketide synthase (PKS) module. The approach was successful, as it identified multiple fusion sites yielding hybrids with better activity than a chimera incorporating 'optimal' junctions previously determined by the same laboratory. As AT exchange is one of the leading strategies for generating novel PKSs and polyketide derivatives, and the approach could be applied to additional functional domains, this work will likely be of high interest to the PKS community.

Nonetheless, there are a number of points which should be addressed prior to acceptance of the manuscript by Nat. Commun. These are detailed below.

Major comments:

1. Nowhere do the authors state directly how the newly-identified sites differ from those previously described as optimal (ref. 6), nor the basis on which optimality was previously predicted/judged. They only refer obliquely to their own research (page 2, line 62 and page 10, line 353), but this comparison should be made explicit in the text, and should be accompanied (in the Supplementary at least) by a figure.
2. Although the authors state the interest of 'normalizing biosensor activation to the amount of heterologous protein produced', the fact that the mCherry signal doesn't correlate directly with solubility but rather with total protein, would appear to negate this argument. Indeed, the overall utility of the calculated 'solubility coefficient' isn't clear, as solubility judged on the basis of the GFP fluorescence alone identifies the same hits as the more elaborate solubility coefficient calculated using the mCherry signal (compare Fig. 4a (highest coefficient values = hits) and Supplementary Fig. 1 (lowest GFP fluorescence = hits)). This is an important point, as adoption by the community would be facilitated by the simpler measure.
3. Fig. 1 c and d. The combined soluble + insoluble fractions for both DEBS M6 cherry and D1 mCherry appear to exceed the total protein. Is there an explanation for this observation?
4. Supplementary Figs. 2-4 are mentioned only once in the text with essentially no explanation (page 5, line 156), and the provided legends are insufficient to allow the reader to understand why the experiments were carried out as well as to interpret the data.

5. Page 9, line 285. The text states that 72 constructs were made, and yet there are many more data points presented in Fig. 5b. Is this due to the fact that 'data points in regions of homology that give identical polypeptide sequence are repeated'? Either an error has been made concerning the number of constructs, or an improved explanation for the lack of accord with the number of presented data points needs to be made here (same comment for page 10, line 304).

6. It would seem that the data listed in the 'Data availability' section fall short of the requirements of the journal. For example, the raw FACS and LC-MS data could/should also be made accessible.

More minor issues:

1. The statement in the abstract (line 29) that PKS hybrids are 'often insoluble due to misfolding' is perhaps an overstatement, as to this reviewer's knowledge at least, this property has not been systematically evaluated in the majority of cases (ref. 6 being a notable exception).

2. In the introduction (page 2, line 45), it would be appropriate to reference additional reviews beyond ref. 1, which specifically cover the PKS area.

3. Page 2, line 56. A reference or references should be cited here to reviews covering PKS genetic engineering.

4. Page 2, line 65. Refs. 8 and 9 are not appropriate here, as they describe the AT specificity within native systems, but not attempts to manipulate it, as for the other citations.

5. Page 2, line 73. To give a representative view of the field, the authors should immediately cite the three publications reporting structures of intact modules (refs. 14–16, but additionally doi: 10.1126/science.abi8358, which was published contemporaneously with 14, and which is mentioned in the Supplementary).

6. Page 2, line 76. The argument that is being made here will not be clear to the non-expert reader – i.e. that the large architectural differences observed in the structures of the PKS modules characterized to date, and notably in terms of the positioning of the linkers flanking the AT domain, do not help to resolve the issue of where the 'optimal' fusion sites may lie.

7. Page 3, line 87. The authors must reference prior art in the field at this stage (i.e. the idea of using solubility as a proxy for function – refs. 20–23), because this idea is by now well-established.

8. Page 3, starting line 100. For the reader unfamiliar with this type of assay, it would be useful to state explicitly that the GFP-based experiment specifically detects misfolded proteins, as the expectation might be that the screen instead identifies good solubility.

9. Page 3, line 109. 'Previously' should be inserted in between two and engineered, to make it clear that reference is being made to earlier work. Also, the epothilone PKS proteins are abbreviated Epo not Epos (see e.g., DOI: 10.1021/bi020006w). (By extension, Tias (page 11) could be Tia)

10. Page 3, line 114. It would be appropriate here to provide a summary statement here concerning how GFP fluorescence was measured.

11. Page 3, line 118. What was the consequence in terms of experimental planning of the observation that the integration of the *gfp* gene into the *ibpA* locus gave a high level of background? (Normally lowercase *gfp* should be used when the gene name is being indicated)
12. Page 4, line 135. The authors could better justify here the choice of mCherry when several fluorescent tags are available (spectroscopic compatibility with GFP, monomeric character, etc.?)
13. Page 4, line 142. Suggest, '...did not substantially change...', as there were in fact minor differences in solubility.
14. Page 4, line 147. The statement that the 'PKSs offer a unique set of challenges unlike smaller proteins', is rather vague. Indeed, whereas the authors expected that the mCherry fluorescence would provide a direct indicator for solubility, they discovered that this wasn't the case, as attachment to an insoluble protein did not apparently diminish the fluorescence. Clearer phrasing would be appropriate here.
15. Page 4, line 151. The authors should make explicit what 'high' IPTG concentrations were relative to 50 μ M, which is presumably low.
16. Page 4, line 155. If the authors decide to present the solubility coefficient, this sentence should be rewritten, as the mCherry fluorescence indicates the overall expression, and the GFP fluorescence, the solubility (or rather the absence of insolubility), respectively.
17. Page 4, line 160. A reference should be included as to the dual specificity of the Epo M4 AT.
18. Page 4, line 163. A brief statement should be included here as to why the TE domain was excluded from the modeling, when the structures of several TE homologs have been solved, and a TE was present in the structure reported in doi: 10.1126/science.abi8358.
19. Fig. 2a. The structure predicted by AlphaFold is symmetrical in the KS-AT region, but asymmetrical in the KR-ACP region, which may come as a surprise to readers unfamiliar with the recently-published high-resolution structures. Perhaps a note of explanation could be added? Perhaps the domains could be color-coded, as the ACP is almost impossible to distinguish.
20. Page 4, line 164. It's not that the predicted 'DEBS M6 structure' showed a high degree of similarity to the solved KS-AT didomain structure, but rather the KS-AT region of the predicted structure.
21. Page 4, line 167. The term 'KS-AT linker' is rather a misnomer, as it has been known since the structures of the first KS-AT didomains were solved, that a portion of the sequence is an independently-folding $\alpha\beta$ adaptor domain, while the flanking regions are linkers in the more classical sense. This fact could be clarified for the non-expert reader. Furthermore, as acknowledged earlier in the text, the precise location/role of the of the post-AT linker depends on which structural model for a PKS module is judged to be correct. (Same comment for page 6, line 213)
22. Fig. 3 legend. At least for this reviewer, the origin of the over-representation of certain swap positions isn't clear from this brief description.
23. Fig. 3c. Can the authors account for the fact that none of the randomly picked colonies contained junctions within the *us80*–*us100* region, although the remaining sequence (*us1*–*us79*) was well represented in this experiment?

24. Page 6, line 200. A sentence or two should be added here explaining which cloning strategy was used to insert the AT library into the parent vector.
25. Page 7, line 221. The word 'removed' doesn't make clear that the proteins were re-expressed in the absence of the mCherry (but in the presence of a His(6?) tag which was used to facilitate their purification).
26. Page 7, line 239. More information should be provided here concerning the in vitro assays, including the character of the 'synthetic starter substrate' (which was a diketide-SNAC of particular stereochemistry, which allowed the authors to assign two of the stereocenters in the final product). It is also of high importance that the authors omitted the NADPH cofactor necessary for the KR domain, thereby ensuring that the product incorporated a C-3 ketone – a feature which facilitated both synthesis of the standard (whose use should be mentioned explicitly here) and product identification by LC-MS.
27. Page 8, line 242. No 'rates' were measured in these experiments, but rather relative overall yields at one or two time points.
28. Page 8, line 250. Strictly speaking, the o15 and o17 variants were not non-functional, but exhibited lower activity.
29. Page 8, line 251. Suggest, '...meaning residues within the post-AT linker which interact with the KS...'
30. Page 8, line 258. The statement here concerning the relative product formation from malonyl- vs. methylmalonyl-CoA should be made quantitative (e.g. for α 4, the amount derived from methylmalonyl-CoA was 23-fold higher)
31. Page 8, line 260. The 'preference of DEBS M6 KS'... for what?
32. Page 8, line 263. The authors should specify here on what basis they chose the 8 and 3 swap junctions. Why wasn't us21 selected, for example, when it gave essentially the same results as us28?
33. Fig. 5. It is odd to place panels c and d next to a, with b below. Also, in the legend to (a) and (b), the authors should indicate that it's the solubility coefficient that's being measured, not the solubility.
34. Page 9, line 288. The authors note the 'surprising' result that 'most other variants showed relatively high solubilities', but fail to put this finding into context with the rest of the results. A comment here would be welcome.
35. Page 9, line 293. It would be useful to indicate which pairs were subjected to further analysis directly in Fig 5b.
36. Supplementary Fig. 5b. In addition to us52/ds25, the purity of us79/ds25 is not ideal either.
37. Page 9, line 296. The one exception noted here should be explicitly identified.
38. Page 9, line 298. Suggest, '...low activities can be attributed to an unfavorable...'
39. Page 9, line 300. The authors cite reference 36 here, but the recent papers reporting more relevant whole module structures also address this issue directly, and so should be mentioned.

40. Page 9, line 316. A reference is needed here to at least one of many existing reviews on PKS genetic engineering.
41. Page 10, line 342. The description of the behavior of the post-AT linker (i.e. 'continues along with the KS domain') could be improved.
42. Page 10, line 346. The authors could profitably add here a statement as to the significance of this particular finding.
43. Methods, cloning and cell cultivation. To permit reproduction of these experiments by the community, substantially more information is required concerning the cloning (i.e. 'some combination of restriction enzyme digestion and Gibson assembly' is far too vague a statement).
44. Methods, Protein fractionation. It would be useful for the non-expert reader to indicate that BAP1 produces phosphopantetheinylated and therefore active recombinant DEBS proteins, a prerequisite for the in vitro assays.
45. Methods, integration of the biosensor into the genome (add 'the'). The authors should provide as Supplementary information, the final sequences of the modified genomic regions (while the full genomes may be available, it will be difficult to find these regions without guidance).
46. Page 12, line 438. Define IDT (Integrated DNA Technology), and provide the company's location.
47. Page 12, line 443. With which enzyme was the plasmid encoding DEBS M6 mCherry digested?
48. Page 13, line 481. The stereochemistry of the diketide should be specified, as well as its origin. An explanation should also be provided for the different time points of analysis of the two assays.
49. The format of the references is not correct (e.g., no abbreviations have been used for the journal names).
50. A list of Supplementary References should be provided in the Supplementary information.

Typographical/grammatical errors (partial list):

1. Page 2, line 51: Suggest '...reduce the newly generated β -keto...' (as such functionality is always generated following chain extension with an acyl-CoA)
2. Page 2, line 75. Suggest, '...it should be possible...'
3. Page 2, line 83. Suggest, '...generate libraries of pikromycin and erythromycin PKS hybrids...'
4. Page 3, line 106. Suggest, '...in frame with the *ibpA* gene...'
5. Page 3, line 113. Suggest, '...were expressed from pET plasmids...'
6. Page 5, line 183. Sentence beginning, 'This was done using...'. Please improve the grammar here.
7. Fig. 3 legend. Please improve the grammar of part (A).
8. Fig. 4 and Supplementary Fig. 5. The symbol font should be used to indicate μM .

9. Page 7, line 238. '...using nickel affinity resin...'
10. Page 8, line 267. 'would' is not the right word here, as the experiment was actually carried out.
11. Fig. 5 legend. 'Induction was 250 μ M IPTG' is not a sentence. In (f) methyl is misspelled.
12. Page 9, line 292. Suggest, '...nearby swap junctions, but with differences...'
13. Page 9, lines 297 and 306. Fig. 5 is being referred to here, not Fig. 6.
14. Page 10, line 360. Suggest, 'Our work suggests that...' (or maybe an even more forceful conclusion would be appropriate here?)
15. Page 11, line 403. Recombinant genes, not recombines genes.
16. Supplementary Fig. 3, 4: upstream is misspelled.
17. Supplementary Table 2. Hygrcin is misspelled.

Reviewer #2 (Remarks to the Author):

Modular polyketide synthases (PKSs) form elaborate quaternary structures to enable the synthesis of natural compounds in a compartmentalized manner. They are responsible for the biosynthesis of a wide range of pharmaceutically relevant polyketide, and have therefore attracted a lot of attention for their technological applications. Up to this date, however, the successes in the rational modulation of polyketide biosynthesis have been moderate, as engineered PKSs often fail to produce the desired compounds. In other cases, the engineered PKSs perform at low rates resulting in low product titers. Among others, a main reason for the slow progress in harnessing PKSs for custom synthesis has for the longest time been the lack of high-throughput methods for their semi-rational engineering. The work of Englund et al. present a fluorescence-based biosensor method as powerful approach for the high-throughput screening of hybrid constructs. They apply this method specifically for replacing the AT domain of DEBS M6 with the AT of EPOS M4, and hypothesize that it will also be applicable to other engineering tasks. They analyze and interpret their results with help of recent structural information about modular PKSs and structural models generated by AI-based structure prediction.

The fluorescence-based biosensor method uses two fluorescent proteins, mCherry and GFP, that are either fused to the protein of interest (mCherry) or under control of a heat-shock inducible promotor. In doing so, the authors can monitor high amounts of mCherry by high intensity of red fluorescence, and low amounts of unfolded protein by low intensity of green GFP fluorescence.

Englund et al. describe a set of experiments organized in logical manner: The authors validate their approach initially with three DEBSM6 constructs, that are well understood in protein quality as reported by the Keasling lab before. They then establish the biosensor assay while creating a new generation of DEBS/EPOS hybrids, which is the main part of this manuscript, and finally harness their insight on DEBS/EPOS hybrid generation in the design of DEBS/TIAS hybrids.

Comments/points requiring revision:

- 1) The authors should share the AlphaFold2 models that were used in this study (and provided for review).
- 2) The authors often refer to “activity”, but strictly speaking just amounts of produced compounds are monitored. Activity does not necessarily correlate with product amounts, given that protein stability of constructs over time may vary. For arguing on activity, turnover rates should be presented.
- 3) On page 2, the authors refer to the differences in structures reported in the studies of Dutta et al. (PikAIII), Bagde et al. (Lsd14) and Cogan et al. (DEBS M2). They argue that “(...) it is still unclear of one could use this structural information to unambiguously predict domain boundaries.” How do the differences in structural models affect domain boundaries? Can the authors be more precise?
- 4) Same paragraph: The authors may want to refer to Cogan et al., when mentioning the “cryo-EM structure on the erythromycin PKS”. Also better say DEBS ...
- 5) Figure 2: Why does the sequence alignment end at us102 and not at us109? This region is still part of the linker.
- 6) Figure legend Figure 2d and e. Please include amino acid numbering in alignments, which helps those readers consulting alignments and structural models that are shared with the readers (see point 1) .
- 7) The figure legend of Fig. 3 says: “Certain swap positions were overrepresented (e.g. ds62) due to gaps in the alignment leading to multiple library variants sharing the same swap junction.” This is difficult to understand, when not explaining better, maybe by referring to the alignment in Fig. 2e.
- 8) Further, the Figure legend says: “Dotted line denotes selected linker region.” Please also explain this in more detail. Can the same dotted lines be inserted in Figure 2d and 2e?
- 9) Legend of Figure 4: The description of figure panel c and d seems to be mixed up; time points 1 hour and 24 hours for methyl...
- 10) Same as point 7 for legend of Figure 5b: “Data points in regions of homology that give identical polypeptide sequence are repeated.” This is again difficult to understand without further explanation.
- 11) Five constructs (o4, o8, o15, o17 and o33) of 40 library colonies were selected to determine solubility by SDS PAGE and enzyme activity by TKL production. It is recommended that for these PKS hybrids the domain boundaries are presented in structural models. This would help understanding the

differences in the domain boundaries and points made in Chapter “In vitro activity assay for highly soluble AT-swapped PKSs”.

12) Figure 5: Solubility data for DEBS M6 wildtype should be added as reference.

13) Page 11: The authors argue that ACP probably docks to with KS-AT linker surface while interacting with the KS for chain elongation, and they refer to Kapur et al. There is recent structural information by Cogan et al. and Bagde et al., both presenting structural data of ACP docked to KS. Rather than referring to Kapur et al., these data should be used for modeling ACP:KS-AT docking and for discussion.

14) In the discussion, the authors refer to a clear correlation of solubility and activity. The correlation is not strict, as for example for D1 being less soluble (Fig. 4a), but producing higher amounts of TKL (4c and d). It is recommended to support the judgement “clear” with a quantitative term. See also point 2 to the use of “activity”.

15) Last paragraph of discussion: The findings of DEBS/EPOS hybrid generation successfully informed the design of DEBS/TIAS hybrid. This is an important result and demonstrates the relevance of the approach. However, these are still too few examples to make a statement about how generalizable the specific results are. It is therefore recommended to tone down the statement.

16) It would be helpful for the non-expert reader to add a scheme to the enzyme mediated synthesis of TKLs.

17) Typo: Fig. 6e, f should be 5e,f.

Reviewer #3 (Remarks to the Author):

In "Biosensor Guided Polyketide Synthase Engineering for Optimization of Domain Exchange Boundaries", Englund et al. apply a promoter responsive to misfolded protein driving expression of a GFP reporter plus a PKS-fused C-terminal mCherry reporter as an overall measure of PKS solubility. They then use this system to determine optimal linker junction locations of the critical AT domain of Type I PKSs to enable the identification of chimeric PKSs that allow the loading of alternative extension substrates. They identified particular swapping regions in both the KS-AT linker and post-AT linker that produced soluble and active proteins, with these regions correlating to those predicted by AlphaFold to be least disruptive of folding, and suggest new design rules that would apply to most diverse PKSs for AT swapping that should minimally impact solubility.

This work is of significance to the natural products and metabolic engineering fields in that it provides a methodology and design guidance for an important aspect of chimeric Type I PKS engineering and using Type I PKSs to access new chemical products. The study appears to be novel and well-conducted, with all proper controls appearing to be present. Some suggestions for enhancing the ability to reproduce this work are provided in line comments pertaining to the Methods section below.

My predominant concern with this manuscript is the relatively limited proof of its generality and universality to PKS engineering as implied in the abstract, with only one base PKS (the most well-studied DEBSM6) and two different ATs (EposM4 and TiasM4) experimentally validated for solubility and desmethyl TKL production. Furthermore, despite the enhanced solubility which did appear to correlate with activity, the activity of the chimeric PKSs for producing desmethyl TKL is still much lower than for the native extension substrate. It is an interesting study however, presenting an incremental advance in being able to more efficiently design and select for active PKSs with alternative activities.

Specific suggested edits and points to address:

Line 76-78: Confusing sentence, please clarify what is meant.

Line 79: Can the authors comment on whether domain boundaries are truly the only issue with designing chimeric PKSs with alternative AT domains? Can't misfolding happen in a manner that is also impossible to predict, or is there extensive evidence (as well as beyond just DEBSM6) that the domain boundary will always be the primary issue?

Line 99: Translational sensors have been applied to Type I PKSs and the reference is missed here - see Mendez-Perez et al., "A translation-coupling DNA cassette for monitoring protein translation in *Escherichia coli*", *Metab. Eng.*, 2012.

Line 182: The statement "one random upstream and downstream junction" was confusing as to what it meant (without looking at the supplementary .xlsx file), clarify e.g. as "random position of the upstream and downstream swap junctions". May also want to define what is meant exactly by "swap junction" somewhere since it is not general terminology as far as I'm aware.

Line 250-255: What are the implications of these interactions for AT swapping of alternative PKSs that are not as well studied as DEBSM6?

Line 259: Is the lower amount of desmethyl TKS equivalent to or worse than alternative products (or the same product with DEBSM6) obtained with Type I PKS AT swaps in other examples in the literature? I'm lacking some context as to how difficult it is to identify a chimeric PKS with 1/10th the activity with a new extension substrate vs the native extension substrate. A major concern here is that the activity will still be inherently limited and identifying highly soluble variants will only help to a certain and perhaps minor extent. The authors seem to hint at some of the issues in both the statement on lines 259-260 about the preference of DEBSM6 KS and lines 300-301 (see below). So does identifying soluble variants help all that much? The activity differences between the selected library variants that are active at all are relatively minimal in Fig 4d.

Line 299-301: This sentence seems to be saying something different about the reduced activity vs what is said on lines 259-260, can the authors clarify? I also found this sentence difficult to interpret.

Line 306: change Fig. 6b, d to Fig. 5b, d.

Line 310 (Discussion section): Can the authors add any suggestions on how they would approach further improving activities of chimeric PKSs? Furthermore, can the approach taken here with the AT domain be used with any other PKS domains or is there a reason it can only be employed with the AT domain? I imagine there will be a huge range of substrate mismatches with pretty much all other domains.

Fig 4a/d: Was desmethyl TKS production measured in any other library variants? I think the correlation between solubility and activity is a bit of a stretch without production measurements from more variants from Fig 4a.

Line 367-368: Very vague description of cloning - which plasmids were restriction digested (and with what enzyme(s)) and which plasmids were constructed by Gibson assembly? Include oligonucleotide sequences in supplementary info for ability to reproduce.

Line 374: change "1% of cultures were inoculated into fresh LB" to "fresh LB was inoculated 1:100 with overnight cultures by volume" or similar.

Line 395: provide oligonucleotide sequences in supplementary info

Line 396: change "Babe" to "Baba".

Line 402: change "chemically BL21" to "chemically competent BL21".

Line 403: change "recombines" to "recombinase".

REVIEWER COMMENTS

Reviewer #1

Review of Biosensor Guided Polyketide Synthase Engineering for Optimization of Domain Exchange Boundaries, by Englund, et al.

This paper reports the use of solubility-guided screening to identify improved sites for swapping of acyl transferase (AT) domains into a model modular polyketide synthase (PKS) module. The approach was successful, as it identified multiple fusion sites yielding hybrids with better activity than a chimera incorporating 'optimal' junctions previously determined by the same laboratory. As AT exchange is one of the leading strategies for generating novel PKSs and polyketide derivatives, and the approach could be applied to additional functional domains, this work will likely be of high interest to the PKS community.

Nonetheless, there are a number of points which should be addressed prior to acceptance of the manuscript by Nat. Commun. These are detailed below.

Major comments:

1. Nowhere do the authors state directly how the newly-identified sites differ from those previously described as optimal (ref. 6), nor the basis on which optimality was previously predicted/judged. They only refer obliquely to their own research (page 2, line 62 and page 10, line 353), but this comparison should be made explicit in the text, and should be accompanied (in the Supplementary at least) by a figure.

We have made a new Supplementary Fig. 6 to show the locations of junctions for the most important variants, including D1 that has the junctions from ref. 6. We have also clearly stated in the discussion how the previous junctions were decided:

"In contrast to our previous suggestion for the optimal domain junction (= us1/ds44), which was determined by *in vitro* kinetic analysis of a series of AT-exchanged PKSs⁸, the current findings indicate the position of domain boundaries in the KS-AT linker can be somewhat flexible over a range of positions."

We have also added to the Discussion section how the newly identified junctions differ from the previously proposed junctions:

"...we revise our previously suggested boundaries for AT domain exchange (= us1/ds44). The above observations suggest that i) when the ds44 junction is selected us3-us17, not us1, should be selected as upstream junctions and ii) when the ds25 junction is selected us3-us17 or us84-us102 should be selected as upstream junctions. In many cases, attempting multiple junctions may not be feasible, and in such instances, we recommend using us3/ds44 (Supplementary Fig. 6)"

2. Although the authors state the interest of 'normalizing biosensor activation to the amount of heterologous protein produced', the fact that the mCherry signal doesn't correlate directly with solubility but rather with total protein, would appear to negate this argument. Indeed, the overall utility of the calculated 'solubility coefficient' isn't clear, as solubility judged on the basis of the GFP fluorescence alone identifies the same hits as the more elaborate solubility coefficient calculated using the mCherry signal (compare Fig. 4a (highest coefficient values = hits) and Supplementary Fig. 1

(lowest GFP fluorescence = hits)). This is an important point, as adoption by the community would be facilitated by the simpler measure.

Yes, it's true that the mCherry values were fairly consistent between variants. In essence therefore, we are comparing differences in GFP when we compare samples. However, we included the mCherry as an internal standard for protein expression because for some PKSs, like for DEBSM6, D0 and D1 in Fig. 1c, there was a difference in expression. Furthermore, if we didn't have the mCherry data, we couldn't be certain that all variants were expressing equally and that the differences in biosensor activation were due to differences in solubility and not from expression.

3. Fig. 1 c and d. The combined soluble + insoluble fractions for both DEBS M6 cherry and D1 mCherry appear to exceed the total protein. Is there an explanation for this observation?

It's unclear to us why the soluble and insoluble fractions add up to more than the total fraction, both when measuring fluorescence and proteins on an SDS-PAGE. It might be due to small errors in the method when separating proteins into different fractions. In the method, lysed cells are centrifuged and the supernatant containing the soluble fraction is removed, so that only the pellet with the insoluble fraction remains. A small amount of the supernatant might still be stuck to the wall of the tube and contaminate the insoluble fraction. However, as these measuring errors are relatively small, we don't think they fundamentally change the results of the experiment.

4. Supplementary Figs. 2-4 are mentioned only once in the text with essentially no explanation (page 5, line 156), and the provided legends are insufficient to allow the reader to understand why the experiments were carried out as well as to interpret the data.

We have expanded our explanation of what those supplementary figures show. The text now reads:

"For each figure in this report where the solubility coefficient is used, a corresponding figure can be found in the supplementary information that presents GFP and mCherry values separately (Supplementary Fig. 1-4)."

We have also changed figure legends for Supplementary Fig. 1-4 to clarify what figure they show the data from.

5. Page 9, line 285. The text states that 72 constructs were made, and yet there are many more data points presented in Fig. 5b. Is this due to the fact that 'data points in regions of homology that give identical polypeptide sequence are repeated'? Either an error has been made concerning the number of constructs, or an improved explanation for the lack of accord with the number of presented data points needs to be made here (same comment for page 10, line 304).

We have changed our text to better explain how 72 constructs could cover the entire sequence:

"The reason why 72 constructs could cover all unique junctions in a 102-amino acid long sequence is due to homology between DEBSM6 and EpoM4 linker sequence.

Junction positions us97-us100, for example, result in the same APGA sequence (Fig. 2d). Therefore, only one construct was made per non-unique junction."

We also changed the text in Fig. 5 to now read:

"Data points from neighboring junction positions that give identical polypeptide sequence due to homology between the parental PKS sequence and exchanged AT sequences are repeated"

6. It would seem that the data listed in the 'Data availability' section fall short of the requirements of the journal. For example, the raw FACS and LC-MS data could/should also be made accessible.

To make our data easily available, we have now included a Supplementary File 4 with all original data for each figure. We also added a new Supplementary Fig. 8 that shows a representative result from the LC-MS measurement. Supplementary Fig. 11 shows how the data measurement from the flow cytometry looked like. As we measured fluorescence of thousands of samples, we only included one representative sample.

More minor issues:

1. The statement in the abstract (line 29) that PKS hybrids are 'often insoluble due to misfolding' is perhaps an overstatement, as to this reviewer's knowledge at least, this property has not been systematically evaluated in the majority of cases (ref. 6 being a notable exception).

We have rewritten parts of the abstract. The sentence in question now reads:

"However, inserting heterologous domains often destabilize PKSs, causing loss of activity and protein misfolding"

By our use of the word "often" in this context, we mean that it's not an uncommon occurrence, but not necessarily something that happens in a majority of cases.

We agree that it's difficult to know how common it is that domain exchanges lead to disruptions of PKS structures. The most comprehensive study of that is likely this study and we did find in Fig. 5b that many variants had lowered solubility coefficient. Furthermore, there is likely an element of survivor bias if we only look at published reports of chimeric PKSs, since engineering attempts that result in misfolded and inactive PKSs are most likely not published.

2. In the introduction (page 2, line 45), it would be appropriate to reference additional reviews beyond ref. 1, which specifically cover the PKS area.

We have added Katz & Baltz 2016 as an additional reference. The article covers pharmaceuticals made from PKSs and has a list of PKS-derived compounds.

3. Page 2, line 56. A reference or references should be cited here to reviews covering PKS genetic engineering.

We added a reference to Weissman 2015.

4. Page 2, line 65. Refs. 8 and 9 are not appropriate here, as they describe the AT specificity within native systems, but not attempts to manipulate it, as for the other citations.

The references were changed to Menzella 2005 for linear chains and Marsden 1998 for branched chains.

5. Page 2, line 73. To give a representative view of the field, the authors should immediately cite the three publications reporting structures of intact modules (refs. 14–16, but additionally doi: 10.1126/science.abi8358, which was published contemporaneously with 14, and which is mentioned in the Supplementary).

By mistake, Bagde et al. was quoted twice instead of Cogan et al. in the second sentence. This error has been corrected.

6. Page 2, line 76. The argument that is being made here will not be clear to the non-expert reader – i.e. that the large architectural differences observed in the structures of the PKS modules characterized to date, and notably in terms of the positioning of the linkers flanking the AT domain, do not help to resolve the issue of where the ‘optimal’ fusion sites may lie.

We changed the text to convey this point more clearly:

“Using this structural information as well as other PKS module structures proposed by cryo-electron microscopy analysis^{17,18}, it could be possible to roughly obtain domain boundary information. However, to unambiguously predict domain boundaries, we believe that experimental validation of domain exchanged PKS libraries is still necessary.”

7. Page 3, line 87. The authors must reference prior art in the field at this stage (i.e. the idea of using solubility as a proxy for function – refs. 20–23), because this idea is by now well-established.

We added: “Using a previously described solubility biosensor” to the introduction to clarify that the use of the biosensor is not novel, only its use for PKS engineering. We also added the references Zutz 2021 and Kraft 2007.

8. Page 3, starting line 100. For the reader unfamiliar with this type of assay, it would be useful to state explicitly that the GFP-based experiment specifically detects misfolded proteins, as the expectation might be that the screen instead identifies good solubility.

We added this sentence when discussing the solubility coefficient to clarify that the biosensor measures insolubility:

“To simplify the presentation of solubility data, we define a solubility coefficient as mCherry fluorescence divided by GFP fluorescence. This coefficient measures how well a PKS is expressed relative to how much it misfolds and activates the biosensor”

We also changed the title of Fig. 1 legend to “Activity of misfolded protein biosensor...” And we added the sentence to the results: “To assess how these misfolded protein biosensors react to PKSs with different levels of solubility”

9. Page 3, line 109. 'Previously' should be inserted in between two and engineered, to make it clear that reference is being made to earlier work. Also, the epothilone PKS proteins are abbreviated Epo not Epos (see e.g., DOI: 10.1021/bi020006w). (By extension, Tias (page 11) could be Tia)

The text has been changed as suggested.

10. Page 3, line 114. It would be appropriate here to provide a summary statement here concerning how GFP fluorescence was measured.

We have added "and fluorescence was measured using a microplate spectrophotometer" and "Around 800 colonies with a visible red color were picked and grown in 96-well plates, and fluorescence was measured using flow cytometry"

We also clarified in Fig. 1 legend what experiment was measured with which method.

11. Page 3, line 118. What was the consequence in terms of experimental planning of the observation that the integration of the *gfp* gene into the *ibpA* locus gave a high level of background? (Normally lowercase *gfp* should be used when the gene name is being indicated)

We are unsure what the reviewer means by "consequence in term of experimental planning" but we added this sentence to emphasize which one of the biosensor strains was selected:

"Of the three tested strains, $\Delta arsB::P_{ibp}$ GFP showed the most promising features as it had low leakiness and almost no activation by the most soluble PKS. Therefore, it was selected for later use."

12. Page 4, line 135. The authors could better justify here the choice of mCherry when several fluorescent tags are available (spectroscopic compatibility with GFP, monomeric character, etc.?)

Unfortunately, we don't have a better reason than it was easily available in the lab and that it worked well when we tested it.

13. Page 4, line 142. Suggest, '...did not substantially change...', as there were in fact minor differences in solubility.

We changed the sentence to now read:

"We observed that mCherry fused to DEBSM6 did not significantly change the solubility"

14. Page 4, line 147. The statement that the 'PKSs offer a unique set of challenges unlike smaller proteins', is rather vague. Indeed, whereas the authors expected that the mCherry fluorescence would provide a direct indicator for solubility, they discovered that this wasn't the case, as attachment to an insoluble protein did not apparently diminish the fluorescence. Clearer phrasing would be appropriate here.

We changed the text to “These results show no reduction in fluorescence depending on PKS solubility, in contrast to what has previously been observed for smaller proteins^{24,29}” to more clearly point out the difference between these results and the results from small proteins.

15. Page 4, line 151. The authors should make explicit what ‘high’ IPTG concentrations were relative to 50 μ M, which is presumably low.

We changed the sentence to:

"Even at high IPTG concentrations (0.25 – 1 mM), DEBSM6 mCherry only weakly induced the biosensor while D0 mCherry had high induction even at 50 μ M (Fig. 1e)"

16. Page 4, line 155. If the authors decide to present the solubility coefficient, this sentence should be rewritten, as the mCherry fluorescence indicates the overall expression, and the GFP fluorescence, the solubility (or rather the absence of insolubility), respectively.

We rewrote the sentence to read:

"To simplify the presentation of solubility data, we define a solubility coefficient as mCherry fluorescence divided by GFP fluorescence. This coefficient measures how well a PKS is expressed relative to how much it misfolds and activates the biosensor"

17. Page 4, line 160. A reference should be included as to the dual specificity of the Epo M4 AT.

We have added the reference Yuzawa et al. 2017 to that sentence.

18. Page 4, line 163. A brief statement should be included here as to why the TE domain was excluded from the modeling, when the structures of several TE homologs have been solved, and a TE was present in the structure reported in doi: 10.1126/science.abi8358.

Although there are structures of TE domains, AlphaFold could not predict the structure when it was attached to DEBSM6. The resulting structure made the TE domain look like a bowl of spaghetti. As our primary interest was with the linker domains flanking the AT, we decided to work with a model without the TE. We changed the text to clarify why we excluded the TE:

"The AlphaFold results from the full DEBSM6 module could not predict the position and structure of the TE domain. Therefore, a model that excluded the TE domain was used."

19. Fig. 2a. The structure predicted by AlphaFold is symmetrical in the KS-AT region, but asymmetrical in the KR-ACP region, which may come as a surprise to readers unfamiliar with the recently-published high-resolution structures. Perhaps a note of explanation could be added? Perhaps the domains could be color-coded, as the ACP is almost impossible to distinguish.

We appreciate the suggestion. We added the following line:

"Promisingly, the generated model contains symmetrical dimeric KS-AT and asymmetrical dimeric KR-ACP, consistent with what was only recently reported from full module structures"

We also changed Fig. 2a so that the ACP text doesn't block the structure and is easier to see. A new Supplementary Fig. 5 has been made that shows the front and back of the structure with domains color coded.

20. Page 4, line 164. It's not that the predicted 'DEBS M6 structure' showed a high degree of similarity to the solved KS-AT didomain structure, but rather the KS-AT region of the predicted structure.

We agree with the reviewer's point and made changes to the text:

"The KS-AT domain of the generated DEBSM6 model showed a high degree of similarity with the experimentally solved structures of KS-AT didomains from DEBSM3 and DEBSM5..."

21. Page 4, line 167. The term 'KS-AT linker' is rather a misnomer, as it has been known since the structures of the first KS-AT didomains were solved, that a portion of the sequence is an independently-folding $\alpha\beta$ adaptor domain, while the flanking regions are linkers in the more classical sense. This fact could be clarified for the non-expert reader. Furthermore, as acknowledged earlier in the text, the precise location/role of the post-AT linker depends on which structural model for a PKS module is judged to be correct. (Same comment for page 6, line 213)

We agree with the reviewer that the term 'KS-AT linker' is a misnomer because it has a tertiary structure. However, in this field, some researchers still use this terminology. To better describe the KS-AT linker structure and function, we added the following two sentences:

"The highly ordered structure of the KS-AT linker suggested that it plays a more significant functional role than merely connecting two domains. In fact, interactions between the KS-AT linker structure and the ACP have been observed during chain elongation."

22. Fig. 3 legend. At least for this reviewer, the origin of the over-representation of certain swap positions isn't clear from this brief description.

The explanation has been expanded to now read:

"Certain junction positions (e.g. ds62) were overrepresented due to how the library was designed: An alignment of DEBSM6-EpoM4 was used to decide junctions. Where there are gaps in the alignment (see position ds62 in Fig. 2e), the same start position of DEBSM6 was used for several end positions of EpoM4. This led to 8 unique variants all sharing ds62 as a junction."

23. Fig. 3c. Can the authors account for the fact that none of the randomly picked colonies contained junctions within the us80-us100 region, although the remaining sequence (us1-us79) was well represented in this experiment?

We think the reason is due to biases when the library was made. As the construction of the library was done using PCR, certain oligos could form secondary structures that

reduced their chances that they would be incorporated during the cloning of the plasmid.

We added a sentence to clarify that the absence of us80-102 was not by design:
"Interestingly, while the reference set showed a mostly even distribution of junctions, except for positions us80-us102 which appeared less frequent than expected,..." "

24. Page 6, line 200. A sentence or two should be added here explaining which cloning strategy was used to insert the AT library into the parent vector.

We added the following explanation:

"After designing and synthesizing the oligo pool library, we amplified EpoAT using PCR with the oligos with KS-AT junctions as "forward" primers and post-AT oligos as "reverse" primers (as shown in Fig. 3a). The resulting fragment mixture was then cloned into the AT position of DEBSM6 mCherry via Gibson assembly"

25. Page 7, line 221. The word 'removed' doesn't make clear that the proteins were re-expressed in the absence of the mCherry (but in the presence of a His(6?) tag which was used to facilitate their purification).

We change the formulation of the sentence to:

"To investigate the solubility of these proteins in the absence of mCherry, plasmids were constructed expressing the variants without the C-terminal fusion tag, and protein amounts in the soluble and insoluble fractions were quantified by SDS-PAGE"

26. Page 7, line 239. More information should be provided here concerning the in vitro assays, including the character of the 'synthetic starter substrate' (which was a diketide-SNAC of particular stereochemistry, which allowed the authors to assign two of the stereocenters in the final product). It is also of high importance that the authors omitted the NADPH cofactor necessary for the KR domain, thereby ensuring that the product incorporated a C-3 ketone – a feature which facilitated both synthesis of the standard (whose use should be mentioned explicitly here) and product identification by LC-MS.

To clarify those points, we made the following changes:

We added a scheme showing the reaction in Fig 4c.

We added information about the stereochemistry of the starter in the methods section:

"2 mM (2S,3R)-3-hydroxy-2-methylpentanoyl-S-N-acetylcyste- amine thioester"

This sentence was added:

"To simplify the analysis, NADPH was omitted so that only the nonreduced product was made instead of a mixture of reduced and nonreduced product."

27. Page 8, line 242. No 'rates' were measured in these experiments, but rather relative overall yields at one or two time points.

We changed the text to:

"in titers similar to the wild-type DEBSM6"

28. Page 8, line 250. Strictly speaking, the o15 and o17 variants were not non-functional, but exhibited lower activity.

We changed the text to:

"Both o15 and o17 had high solubility but showed significantly lower production"

29. Page 8, line 251. Suggest, '...meaning residues within the post-AT linker which interact with the KS...'

We appreciate the suggestion and have changed the text accordingly.

30. Page 8, line 258. The statement here concerning the relative product formation from malonyl- vs. methylmalonyl-CoA should be made quantitative (e.g. for $\alpha 4$, the amount derived from methylmalonyl-CoA was 23-fold higher)

We changed the text to:

"In contrast, o4, o8 and o33 all showed higher production than the less soluble reference D1, albeit at a ~23 times lower amount compared with methyl TKL production"

We wrote 23 times as that was the average of o04 (24x), o08 (22x) and o33 (23x).

31. Page 8, line 260. The 'preference of DEBS M6 KS'... for what?

We changed the text to:

"This is consistent with previous results that have shown DEBSM6 to be less accepting of malonyl-CoA compared with the native methylmalonyl-CoA substrate. This is thought to be due to gatekeeping by the rest of the PKS domains e.g. the KS⁸."

32. Page 8, line 263. The authors should specify here on what basis they chose the 8 and 3 swap junctions. Why wasn't us21 selected, for example, when it gave essentially the same results as us28?

Our goal was to select evenly spaced-out junctions with different effects on solubility.

We changed the sentence to:

"Next, we selected eight junction positions that were evenly spaced out in the KS-AT linker and three in the post-AT linker and constructed all 24 combinations"

33. Fig. 5. It is odd to place panels c and d next to a, with b below. Also, in the legend to (a) and (b), the authors should indicate that it's the solubility coefficient that's being measured, not the solubility.

It is a bit odd, but we couldn't find a different solution. Fig. 5b requires a whole page width and 5a needs to come before 5b.

We changed the figure legend for (a) and (b) so that it states that solubility coefficient was measured and not solubility.

34. Page 9, line 288. The authors note the 'surprising' result that 'most other variants showed relatively high solubilities', but fail to put this finding into context with the rest of the results. A comment here would be welcome.

We have added text to the Discussion section to discuss those findings:

"In our detailed investigation of junctions in the DEBS KS-AT linker, we observed that most variants are highly soluble except for the ones that have the upstream junctions between $\alpha 2$ and $\beta 2$ structure regions, which are less soluble and less active (us53 and us79). In contrast to our previous suggestion for the optimal domain junction (= us1/ds44), which was determined by in vitro kinetic analysis of a series of AT-exchanged PKSs⁸, the current findings indicate the position of domain boundaries in KS-AT linker can be somewhat flexible over a range of positions."

35. Page 9, line 293. It would be useful to indicate which pairs were subjected to further analysis directly in Fig 5b.

Fig. 5b was changed as suggested to show what pairs were used for further analysis. Fig. 5e has also been changed to remove the solubility data as it's no longer needed.

36. Supplementary Fig. 5b. In addition to us52/ds25, the purity of us79/ds25 is not ideal either.

We modified the text to read:

"These PKSs were purified without the mCherry fusion tag, and all proteins were isolated with high purity except us52/ds25 and to a lesser extent us64/ds25 (Supplementary Fig. 7b)."

37. Page 9, line 296. The one exception noted here should be explicitly identified.

We modified the text to read:

"*In vitro* analysis of desmethyl and the methyl TKL production showed that the less soluble mutant pair made less product in all examples but one, and these results were consistent for both TKL products (Fig. 5e, f). The one exception was us27 that had the junction in the aforementioned $\beta 1$ proline."

38. Page 9, line 298. Suggest, '...low activities can be attributed to an unfavorable...'

We thank the reviewer for the suggestion and modified the text accordingly:

"Because all tested variants have the same post-AT linker junction (ds25), the cause of the structural destabilization and low production can be attributed to an unfavorable junction in the KS-AT linker"

39. Page 9, line 300. The authors cite reference 36 here, but the recent papers reporting more relevant whole module structures also address this issue directly, and so should be mentioned.

We rearranged the arguments within the text and this part is not present anymore.

40. Page 9, line 316. A reference is needed here to at least one of many existing reviews on PKS genetic engineering.

We added the reference to Barajas 2017.

41. Page 10, line 342. The description of the behavior of the post-AT linker (i.e. 'continues along with the KS domain') could be improved.

We have rephrased the sentence:

“For the post-AT linker, all predicted modules exit the AT domain with α_4 , after which the linkers wrap around KS-AT linkers and continues alongside the KS domains where LPTY(A/P)FQ(H/R)xRYWL motif binds to the KS surface after which the models diverge depending on what domain that follows.”

42. Page 10, line 346. The authors could profitably add here a statement as to the significance of this particular finding.

See our reply to comment 34.

43. Methods, cloning and cell cultivation. To permit reproduction of these experiments by the community, substantially more information is required concerning the cloning (i.e. ‘some combination of restriction enzyme digestion and Gibson assembly’ is far too vague a statement).

As we constructed ~170 chimeric PKSs for this study, it's difficult to in detail explain how each was made. We believe that as long as the plasmid sequence is available, then other researchers can replicate this study. Therefore, we have made the sequences available for the most important plasmids listed in Supplementary Table 3, and there is information there on how to order a physical copy of the plasmids. The rest (the plasmids constructed for Fig. 5b) can be reconstructed with the sequence of DEBS mCherry (available in Supplementary Table 3) and the information from Fig 5b.

44. Methods, Protein fractionation. It would be useful for the non-expert reader to indicate that BAP1 produces phosphopantetheinylated and therefore active recombinant DEBS proteins, a prerequisite for the in vitro assays.

We modified the text to read:

“The *E. coli* strain BAP1⁴⁶ was used for expression of proteins. BAP1 is based on BL21 (DE3) and expresses a phosphopantetheinyl transferases that is necessary for purifying PKSs in their holo-form”

45. Methods, integration of the biosensor into the genome (add ‘the’). The authors should provide as Supplementary information, the final sequences of the modified genomic regions (while the full genomes may be available, it will be difficult to find these regions without guidance).

We have included the sequences of the three biosensors as Supplementary Files 1.

46. Page 12, line 438. Define IDT (Integrated DNA Technology), and provide the company’s location.

We added the definition of IDT and included the location of the companies.

47. Page 12, line 443. With which enzyme was the plasmid encoding DEBS M6 mCherry digested?

We modified the text to include that information:

"The resulting product was cloned into DEBSM6 mCherry digested with KpnI and BamHI"

48. Page 13, line 481. The stereochemistry of the diketide should be specified, as well as its origin. An explanation should also be provided for the different time points of analysis of the two assays.

We added information about the stereoisomeric form of the starter and where it came from: "2 mM (2S,3R)-3-hydroxy-2-methylpentanoyl-S-N-acetylcyste- amine thioester (synthesized as previously described⁸)..."

We also added the following information about the time points:

"Reactions were incubated at room temperature and samples were taken at 1 hour for the methylmalonyl-CoA reaction and 24 hours for both methyl- and malonyl-CoA samples. Samples were only taken at 24 hours for the reaction with malonyl-CoA due to too low product formation at 1 hour."

49. The format of the references is not correct (e.g., no abbreviations have been used for the journal names).

There was an error in the reference manager that has been fixed. We thank the reviewer for noticing the error.

50. A list of Supplementary References should be provided in the Supplementary information.

A reference list has been added to Supplementary Information.

Typographical/grammatical errors (partial list):

1. Page 2, line 51: Suggest '...reduce the newly generated b-keto...' (as such functionality is always generated following chain extension with an acyl-CoA)
2. Page 2, line 75. Suggest, '...it should be possible...'
3. Page 2, line 83. Suggest, '...generate libraries of pikromycin and erythromycin PKS hybrids...'
4. Page 3, line 106. Suggest, '...in frame with the *ibpA* gene...'
5. Page 3, line 113. Suggest, '...were expressed from pET plasmids...'
6. Page 5, line 183. Sentence beginning, 'This was done using...'. Please improve the grammar here.
7. Fig. 3 legend. Please improve the grammar of part (A).
8. Fig. 4 and Supplementary Fig. 5. The symbol font should be used to indicate μM .
9. Page 7, line 238. '...using nickel affinity resin...'
10. Page 8, line 267. 'would' is not the right word here, as the experiment was actually carried out.
11. Fig. 5 legend. 'Induction was 250 μM IPTG' is not a sentence. In (f) methyl is misspelled.
12. Page 9, line 292. Suggest, '...nearby swap junctions, but with differences...'
13. Page 9, lines 297 and 306. Fig. 5 is being referred to here, not Fig. 6.
14. Page 10, line 360. Suggest, 'Our work suggests that...' (or maybe an even more forceful conclusion would be appropriate here?)
15. Page 11, line 403. Recombinant genes, not recombinates genes.

16. Supplementary Fig. 3, 4: upstream is misspelled.

17. Supplementary Table 2. Hygrcin is misspelled.

We thank the reviewer for their sharp eye and good suggestions. We made the suggested changes to the text.

Reviewer #2:

Modular polyketide synthases (PKSs) form elaborate quaternary structures to enable the synthesis of natural compounds in a compartmentalized manner. They are responsible for the biosynthesis of a wide range of pharmaceutically relevant polyketide, and have therefore attracted a lot of attention for their technological applications. Up to this date, however, the successes in the rational modulation of polyketide biosynthesis have been moderate, as engineered PKSs often fail to produce the desired compounds. In other cases, the engineered PKSs perform at low rates resulting in low product titers. Among others, a main reason for the slow progress in harnessing PKSs for custom synthesis has for the longest time been the lack of high-throughput methods for their semi-rational engineering. The work of Englund et al. present a fluorescence-based biosensor method as powerful approach for the high-throughput screening of hybrid constructs. They apply this method specifically for replacing the AT domain of DEBS M6 with the AT of EPOS M4, and hypothesize that it will also be applicable to other engineering tasks. They analyze and interpret their results with help of recent structural information about modular PKSs and structural models generated by AI-based structure prediction.

The fluorescence-based biosensor method uses two fluorescent proteins, mCherry and GFP, that are either fused to the protein of interest (mCherry) or under control of a heat-shock inducible promoter. In doing so, the authors can monitor high amounts of mCherry by high intensity of red fluorescence, and low amounts of unfolded protein by low intensity of green GFP fluorescence.

Englund et al. describe a set of experiments organized in a logical manner: The authors validate their approach initially with three DEBSM6 constructs, that are well understood in protein quality as reported by the Keasling lab before. They then establish the biosensor assay while creating a new generation of DEBS/EPOS hybrids, which is the main part of this manuscript, and finally harness their insight on DEBS/EPOS hybrid generation in the design of DEBS/TIAS hybrids.

Comments/points requiring revision:

1) The authors should share the AlphaFold2 models that were used in this study (and provided for review).

All models have been made available as pdb:s in Supporting File 3.

2) The authors often refer to “activity”, but strictly speaking just amounts of produced compounds are monitored. Activity does not necessarily correlate with product amounts, given that protein stability of constructs over time may vary. For arguing on activity, turnover rates should be presented.

We changed references to activity throughout the manuscript and we now instead use "productivity".

3) On page 2, the authors refer to the differences in structures reported in the studies of Dutta et al. (PikAIII), Bagde et al. (Lsd14) and Cogan et al. (DEBS M2). They argue that "(...) it is still unclear of one could use this structural information to unambiguously predict domain boundaries." How do the differences in structural models affect domain boundaries? Can the authors be more precise?

Our point is that significant differences in empirically determined structures make it difficult to solely depend on them to determine domain boundaries. These boundaries need to be validated experimentally. To improve our formulation of that idea, we changed the paragraph to:

"Using this structural information as well as other PKS module structures proposed by cryo-electron microscopy analysis^{17,18}, it should be possible to roughly obtain domain boundary information. However, to unambiguously predict domain boundaries, we believe that experimental validation of domain exchanged PKS libraries is still necessary."

4) Same paragraph: The authors may want to refer to Cogan et al., when mentioning the "cryo-EM structure on the erythromycin PKS". Also better say DEBS ...

By mistake we referenced Badger twice instead of Cogan. This error has been corrected.

5) Figure 2: Why does the sequence alignment end at us102 and not at us109? This region is still part of the linker.

Due to limitations of the length of the oligos in the oligo pool libraries, the entire linker region was not covered. We have added this information to the text:

"Due to length limitations on the oligos, the entire KS-AT linker region could not be selected and the ten positions closest to the AT domain were excluded (Fig. 2b, us103-us112)."

Also, by using a different method to predict where the AT domain starts, to be consistent with the method used in Supplementary Fig. 10, the start of the AT has been changed from us109 to us112.

6) Figure legend Figure 2d and e. Please include amino acid numbering in alignments, which helps those readers consulting alignments and structural models that are shared with the readers (see point 1).

We added amino acid numbering to Fig. 2 legend:

"For reference, the position of the first amino acid in the KS-AT sequence is I1908 for DEBSM6 and V1948 for EpoM4 and in the post-AT linker, the first position is A2301 and P2345 respectively."

7) The figure legend of Fig. 3 says: "Certain swap positions were overrepresented (e.g. ds62) due to gaps in the alignment leading to multiple library variants sharing the same swap junction." This is difficult to understand, when not explaining better, maybe by referring to the alignment in Fig. 2e.

See our response to reviewer 1, comment 22.

8) Further, the Figure legend says: "Dotted line denotes selected linker region." Please also explain this in more detail. Can the same dotted lines be inserted in Figure 2d and 2e?

The method we used to construct the junction library used oligonucleotides that had a max length of 350 base pairs. Because of this size limitation, we were restricted to test only a ~100 amino acid long sequence, which did not cover the entire annotated KS-AT linker region. Therefore, we had to select which part of the linker regions we would include in the junction library. These sequences are the "selected linker regions" and were us1-us102 and ds1-ds90. Therefore, the entire sequence in Fig. 2d and e are the selected linker sequences.

We think that our original explanation did not provide enough details to explain how the junction library was made. Therefore, we have added the following new paragraph to the paper:

"Next, we developed an in vitro method for creating the randomized junction library of DEBSM6 engineered with the AT from EpoM4. Each variant of the library was designed to have the upstream and downstream junction randomly positioned somewhere in the KS-AT linker and post-AT linker, respectively (Fig. 3a). To make the library, we used pooled oligonucleotides of up to 350 base pairs in length. The oligo pool library was designed by firstly selecting regions in the KS-AT linker and in the post-AT linker. Each amino acid in these regions corresponded to a specific placement of the DEBSM6-EpoM4 junction and were sequentially named us(upstream)1-102 (Fig. 2d) and ds(downstream)1-90 (Fig. 2e). Due to length limitations on the oligos, the entire KS-AT linker region could not be selected and the ten positions closest to the AT domain were excluded (Fig. 2b, us103-us112). In contrast, the post-AT linker was short enough that sequences outside of the linker region (residues inside the AT and KR domain) were also included in the library (Fig. 2c). Next, we designed one oligo per unique junction position from the selected regions by aligning the DEBSM6 sequence with EpoM4. Although the selected KS-AT linker region was 102 amino acids long, only 72 oligos had unique sequences due to the presence of conserved regions where DEBSM6 and EpoM4 have stretches of identical amino acid sequences. In total, 72 unique oligos were created for the KS-AT linker and 73 for the post-AT linker, thereby resulting in 5,256 possible combinations when randomly combining an upstream and downstream junction."

9) Legend of Figure 4: The description of figure panel c and d seems to be mixed up; time points 1 hour and 24 hours for methyl...

Yes they were in fact mixed up. We thank the reviewer for noticing this error.

10) Same as point 7 for legend of Figure 5b: "Data points in regions of homology that give identical polypeptide sequence are repeated." This is again difficult to understand without further explanation.

Yes, this part of the text could have been explained better. See our answer to reviewer 1, comment 5.

11) Five constructs (o4, o8, o15, o17 and o33) of 40 library colonies were selected to determine solubility by SDS PAGE and enzyme activity by TKL production. It is recommended that for these PKS hybrids the domain boundaries are presented in structural models. This would help understanding the differences in the domain boundaries and points made in Chapter “In vitro activity assay for highly soluble AT-swapped PKSs”.

We agree that it is helpful to be able to compare different junction positions and to see where in the structure model they are. Therefore, we have made a new Supplementary Fig. 6 where the junction positions of several different variants are displayed and where on a structure model the residues are located.

12) Figure 5: Solubility data for DEBS M6 wildtype should be added as reference.

DEBSM6 was measured at the same time as the samples in Figure 5 but was not included in the figure due to it being too far outside the y-axis range. But we added the following information to the figure legend:

“For reference, DEBSM6 mCherry solubility coefficient was 71.1 ± 0.5 .”

13) Page 11: The authors argue that ACP probably docks with KS-AT linker surface while interacting with the KS for chain elongation, and they refer to Kapur et al. There is recent structural information by Cogan et al. and Bagde et al., both presenting structural data of ACP docked to KS. Rather than referring to Kapur et al., these data should be used for modeling ACP:KS-AT docking and for discussion.

That sentence has been moved to earlier in the results. We have changed the references to the ones the reviewer suggested, as well as another reference to Feng et al. 2022 and Kapur et al. 2012. The new text now reads:

“In fact, interactions between the KS-AT linker structure and the ACP have been observed during chain elongation^{16,17,35,36}”

14) In the discussion, the authors refer to a clear correlation of solubility and activity. The correlation is not strict, as for example for D1 being less soluble (Fig. 4a), but producing higher amounts of TKL (4c and d). It is recommended to support the judgment “clear” with a quantitative term. See also point 2 to the use of “activity”.

We agree that the solubility and activity correlation is not strict as there are many other factors that determine activity, and solubility is only a part of it. To make this point more clearly, we have added a section to the discussion on line 426-436. We also removed language overgeneralizing the solubility - activity relationship throughout the paper. For example, we have removed this sentence from the abstract as it claimed it was activity we sorted for and not solubility: “Together, we have successfully developed an experimentally validated high-throughput method to efficiently screen active engineered PKSs that produce target molecules.”

We also added the following text to give a quantitative number on how often we observed a more soluble variant to be more active than a less soluble variant:

“However, our methodology can at least remove structurally disrupted variants from a domain exchanged PKS library. These variants are relevant to remove, as in 3 out of

4 cases, we observed that less soluble variants made less product than nearly sequence identical high solubility variants (Fig. 5e, f)."

15) Last paragraph of discussion: The findings of DEBS/EPOS hybrid generation successfully informed the design of DEBS/TIAS hybrid. This is an important result and demonstrates the relevance of the approach. However, these are still too few examples to make a statement about how generalizable the specific results are. It is therefore recommended to tone down the statement.

The paragraph in question has been changed to tone down the universality of the method. The following sentence has been removed from the start of the paragraph: "In summary, we created a method to easily and rapidly assess solubility of hybrid PKSs, which correlates well with enzyme activity and increases the throughput of screening."

Also, the last sentence of the paragraph has been changed from "Our work may suggest that this system could also be used to select active domain-swapped PKSs with other PKS domains such as KS, KR, DH, and/or ER domains to fully optimize each PKS domain swap junctions." to "Our work suggests that this system could also be used with other PKS domains such as KS, KR, DH, and/or ER domains to fully optimize domain boundaries of domain-exchanged PKSs."

16) It would be helpful for the non-expert reader to add a scheme to the enzyme mediated synthesis of TKLs.

We have added a reaction scheme as Fig. 4c.

17) Typo: Fig. 6e, f should be 5e,f.

We thank the reviewer for noticing this error.

Reviewer #3:

In "Biosensor Guided Polyketide Synthase Engineering for Optimization of Domain Exchange Boundaries", Englund et al. apply a promoter responsive to misfolded protein driving expression of a GFP reporter plus a PKS-fused C-terminal mCherry reporter as an overall measure of PKS solubility. They then use this system to determine optimal linker junction locations of the critical AT domain of Type I PKSs to enable the identification of chimeric PKSs that allow the loading of alternative extension substrates. They identified particular swapping regions in both the KS-AT linker and post-AT linker that produced soluble and active proteins, with these regions correlating to those predicted by AlphaFold to be least disruptive of folding, and suggest new design rules that would apply to most diverse PKSs for AT swapping that should minimally impact solubility.

This work is of significance to the natural products and metabolic engineering fields in that it provides a methodology and design guidance for an important aspect of chimeric Type I PKS engineering and using Type I PKSs to access new chemical products. The study appears to be novel and well-conducted, with all proper controls appearing to

be present. Some suggestions for enhancing the ability to reproduce this work are provided in line comments pertaining to the Methods section below.

My predominant concern with this manuscript is the relatively limited proof of its generality and universality to PKS engineering as implied in the abstract, with only one base PKS (the most well-studied DEBSM6) and two different ATs (EposM4 and TiasM4) experimentally validated for solubility and desmethyl TKL production. Furthermore, despite the enhanced solubility which did appear to correlate with activity, the activity of the chimeric PKSs for producing desmethyl TKL is still much lower than for the native extension substrate. It is an interesting study however, presenting an incremental advance in being able to more efficiently design and select for active PKSs with alternative activities.

One issue with the original manuscript was overstated claims on potential applications of the biosensor. To address this, we have made several changes to the manuscript. We added a section to the discussion where we described limitations to this type of biosensor (line 426-446). We have also removed statements that the biosensor can be used to sort out chimeric PKSs with higher activity, and lessen claims on how general the method is.

In spite of these limitations, we still feel that this manuscript is novel enough to be published in Nature Communications. We constructed around 170 engineered PKSs for this project. From them, we generated Fig. 5b which provides a unique insight into what occurs to the structure of a PKS during domain exchange. We have also updated our domain boundaries recommendation from Yuzawa et al. ACS Synth Biol 2017 (= ref. 8, cited >100 in the last five years), which should be useful for many academic and industrial scientists in this research community. Very few PKS papers have used high throughput methods to engineer PKSs, and none described so far have used biosensors.

Regarding the chimeric PKSs having low production of desmethyl TKLs, there are several issues with domain exchange that need to be overcome before the method can consistently produce active chimeric PKSs. Our focus here is how to optimally insert new functional domains without causing disruption between the parental PKS and the inserted domain. However, the issue that the reviewer brings up here, that the AT exchanged DEBSM6 is less productive in making desmethyl TKLs than its native methyl TKL, will likely require increasing the promiscuity of the KS domain to be more accepting of an unnatural substrate, which we are currently working on. We likely need progress on both how to stably insert new domains and how to make the rest of the PKS accept the new product, for the domain exchange method to be successful.

Specific suggested edits and points to address:

Line 76-78: Confusing sentence, please clarify what is meant.

See our response to reviewer 2, comment 3. The text has been rewritten for clarification.

Line 79: Can the authors comment on whether domain boundaries are truly the only issue with designing chimeric PKSs with alternative AT domains? Can't misfolding

happen in a manner that is also impossible to predict, or is there extensive evidence (as well as beyond just DEBSM6) that the domain boundary will always be the primary issue?

There are many obstacles for successfully exchanging PKS domains. For example, one of the primary issues not related to changes in protein structure is that the other domains can have poor specificities towards the new substrate, in essence acting as gatekeepers. Also, the domain boundaries are not the only factor that influences solubility. The AT domain itself can have properties that make it more or less soluble. You can see that in Fig. 5b where a domain exchange with TiaAT results in higher solubility coefficient than with EpoAT, indicating that the AT domain itself influences solubility regardless of boundaries. The domain boundaries are one factor that can critically influence the function of the PKS, which we have focused on in this report.

We have added the following line to the discussion:

“Furthermore, domain exchange with TiaAT gave, on average, a higher solubility coefficient than when EpoAT was used, indicating that the AT domains themselves influence solubility regardless of which domain boundaries are used.”

Line 99: Translational sensors have been applied to Type I PKSs and the reference is missed here - see Mendez-Perez et al., "A translation-coupling DNA cassette for monitoring protein translation in Escherichia coli", Metab. Eng., 2012.

We thank the reviewer for bringing this article to our attention. We have added the reference to the following sentence:

"Although other methods for monitoring PKS expression have been reported³⁰, the use of fusion tags is unexplored."

Line 182: The statement "one random upstream and downstream junction" was confusing as to what it meant (without looking at the supplementary .xlsx file), clarify e.g. as "random position of the upstream and downstream swap junctions". May also want to define what is meant exactly by "swap junction" somewhere since it is not general terminology as far as I'm aware.

To clarify what we mean with these terms, we have added this part early on in the results section:

"DEBSM6 was previously shown to be more soluble than D1, which in turn was more soluble than D0 due to differences in linker junctions⁸, i.e. the positions in the linker regions where the parental DEBS sequence end and the heterologous AT sequence begins. As the AT domain is typically located in the middle of the PKS gene, PKSs engineered with AT exchanges carry two junctions, one upstream of the AT in the KS-AT linker and one downstream in the post-AT linker."

Additionally, we changed references to "swap junctions", as those words sound a bit too colloquial, to "linker junctions". We also added a section starting on line 203, to explain in greater detail how the randomized junction library was created. We hope with these inclusions that the reader will better understand how the method worked.

Line 250-255: What are the implications of these interactions for AT swapping of alternative PKSs that are not as well studied as DEBSM6?

Several KS-AT didomain structures are available and, in all cases, these post AT linker/KS interactions have been clearly observed. It appears that these interactions are conserved in most PKS modules. We have modified our text to include this point: "These barely active variants had the downstream junction at ds62 and ds64, respectively, meaning residues within the post-AT linker which interacts with the KS (ds44-56 in AlphaFold prediction) had the heterologous EpoM4 AT sequence. This part of the linker is known to tightly interact with the KS in DEBS^{33,34} and in other PKSs^{16,39}, and is critical for KS condensation reaction⁴⁰, indicating that the heterologous linker sequence in the o15/o17 variants is unable to complement that function⁴⁰."

In our reworked discussion, we have added a small discussion where we suggest optimal domain boundaries based on these results. Part of the new text now reads: "In summary, based on in vivo solubility data and in vitro activity data, as well as structure data predicted by AlphaFold2, we revise our previously suggested boundaries for AT domain exchange (= us1/ds44). The above observations suggest that i) when the ds44 junction is selected us3-us17, not us1, should be selected as upstream junctions and ii) when the ds25 junction is selected us3-us17 or us84-us102 should be selected as upstream junctions. In many cases, attempting multiple junctions may not be feasible, and in such instances, we recommend using us3/ds44 (Supplementary Fig. 6)"

Line 259: Is the lower amount of desmethyl TKS equivalent to or worse than alternative products (or the same product with DEBSM6) obtained with Type I PKS AT swaps in other examples in the literature? I'm lacking some context as to how difficult it is to identify a chimeric PKS with 1/10th the activity with a new extension substrate vs the native extension substrate. A major concern here is that the activity will still be inherently limited and identifying highly soluble variants will only help to a certain and perhaps minor extent. The authors seem to hint at some of the issues in both the statement on lines 259-260 about the preference of DEBSM6 KS and lines 300-301 (see below). So does identifying soluble variants help all that much? The activity differences between the selected library variants that are active at all are relatively minimal in Fig 4d.

Our use of the biosensor in this project was to find domain boundaries that disrupts the native protein structure as little as possible. As we mentioned above, issues like gatekeeping is still a big problem that is not being addressed by this project. However, poorly chosen domain boundaries can cripple a chimeric PKS. This is apparent in Fig. 5e, f, where some boundaries almost abolished activity of the PKS. Therefore, we think this is an issue that is worth tackling separate from all other issues.

As mentioned before, we have added a section to discuss the lower amount of desmethyl TKL. See line 427-433.

Line 299-301: This sentence seems to be saying something different about the reduced activity vs what is said on lines 259-260, can the authors clarify? I also found this sentence difficult to interpret.

Line 259-260 discusses why AT-exchanged DEBSM6 has lower production when using its nonnative substrate malonyl-CoA compared with its native substrate methylmalonyl-CoA. This reduction in activity is thought to be due to gatekeeping of the other domains, ie. the KS is less active towards its nonnative substrate. We saw the same reduction in activity in our previous report (Yuzawa et al. 2017).

Line 299-301 discusses why two identical AT-exchanged DEBSM6, except for a few amino acids in the linker region, can have such large differences in amount of production. The variants us48 and us52 from Fig. 5e, for example, has their domain boundaries placed 5 amino acids apart from each other. This small change in the position of the junctions lead to a large difference in solubility and almost a complete abolishment of activity for us52. These two variants highlight a difficulty in PKS engineering and provide an important insight into domain junctions in AT domain exchanges.

Line 306: change Fig. 6b, d to Fig. 5b, d.

We thank the reviewer for bringing this error to our attention

Line 310 (Discussion section): Can the authors add any suggestions on how they would approach further improving activities of chimeric PKSs? Furthermore, can the approach taken here with the AT domain be used with any other PKS domains or is there a reason it can only be employed with the AT domain? I imagine there will be a huge range of substrate mismatches with pretty much all other domains.

To further improve activities of chimeric PKSs, you would most likely have to address the gatekeeping issue that we previously mentioned. We discuss that in our new discussion part:

"Despite using domain boundaries that minimally affected protein stability, the production of the desmethyl product was still >10 times lower than that of the non-native product. Improving domain junctions is unlikely to increase the production of desmethyl TKL further. Instead, increasing the promiscuity of the other domains in the module through point mutations would likely be necessary"

For this project, we have limited ourselves to finding optimal domain boundaries for AT domains. The reason for that was our previous report (Yuzawa 2017) demonstrated that domain boundaries can have a big effect on solubility and activity. However, we see no reason why you couldn't apply the same method for optimizing other domain exchanges such as for the KR. Therefore, we have this sentence at the end of the manuscript:

"Our work suggests that this system could also be used with other PKS domains such as KS, KR, DH, and/or ER domains to fully optimize domain boundaries of domain exchanged PKSs."

Fig 4a/d: Was desmethyl TKS production measured in any other library variants? I think the correlation between solubility and activity is a bit of a stretch without production measurements from more variants from Fig 4a.

Those five were the only ones we measured from the randomized junction library. That data is not enough on its own to show a correlation between solubility and activity.

However, the purpose of Fig. 5 e, f was to further test this correlation and in 3/4 cases, the less soluble variant produced less product and in the last case, there was the same amount of production. Based on these two data sets, we believe that the correlation is strong enough in DEBSM6 to at least say that domain boundaries that maintain protein stability are more likely to generate active proteins than boundaries that are destabilizing. Our previous version of the manuscript overstated the activity - solubility relationship but that has now been changed so that we make less strong claims. See our response to reviewer 2, comment 14.

We have added the following explanation to the discussion:

"However, our methodology can at least remove structurally disrupted variants from a domain exchanged PKS library. These variants are relevant to remove, as in 3 out of 4 cases, we observed that less soluble variants made less product than nearly sequence identical high solubility variants (Fig. 5e, f)."

Line 367-368: Very vague description of cloning - which plasmids were restriction digested (and with what enzyme(s)) and which plasmids were constructed by Gibson assembly? Include oligonucleotide sequences in supplementary info for ability to reproduce.

See our response to reviewer 1, comment 43.

We have added some additional details on the restriction enzymes used in the library generation:

"The resulting product was cloned into DEBSM6 mCherry digested with KpnI and BamHI"

Line 374: change "1% of cultures were inoculated into fresh LB" to "fresh LB was inoculated 1:100 with overnight cultures by volume" or similar.

We changed the text according to the reviewer's suggestion.

Line 395: provide oligonucleotide sequences in supplementary info

See our answer above to the question from reviewer 1, comment 43. We have made sequences of plasmids available in Supplementary Table 3, the oligo sequences from making the randomized junction library are available as Supplementary File 2, and the DNA sequences of the biosensors are available as Supplementary File 1.

Line 396: change "Babe" to "Baba".

Line 402: change "chemically BL21" to "chemically competent BL21".

Line 403: change "recombines" to "recombinase".

We thank the reviewer for bringing these errors to our attention. We have changed them as suggested.

REVIEWERS' COMMENTS

Reviewer #1 (Remarks to the Author):

The authors have successfully addressed the majority of my comments and those of reviewer 2.

I do have a couple of additional, minor comments/suggestions:

1. It is perhaps unwarranted to state in the Abstract that the identified junctions are 'optimal', given that in previous work, the author also claimed to have identified 'optimal' fusion points which they now revise (see line 417 in the revised document). Indeed they use the term 'fully optimize' in the discussion (line 459), which implies there are degrees of optimization.
2. Response to comment 6: the authors included 'a representative result from LC-MS measurement', but all of the raw data files should be made available, as is standard for the journal.
2. Text written in response to comment 23: 'frequent' should be 'frequently'
3. Text written in response to comment 41: 'the linkers wrap around KS-AT linkers and continues...' The grammar here should be corrected.
4. Text, line 382. It is too precise to state that 'a hundred natural polyketide analogs have been generated', as to my knowledge, no one has published an exact count.

Reviewer #3 (Remarks to the Author):

I thank the authors for their detailed responses and edits and believe that all concerns and errors were adequately addressed in the revised manuscript. I therefore recommend the revised manuscript for publication.

REVIEWERS' COMMENTS

Reviewer #1 (Remarks to the Author):

The authors have successfully addressed the majority of my comments and those of reviewer 2.

I do have a couple of additional, minor comments/suggestions:

1. It is perhaps unwarranted to state in the Abstract that the identified junctions are 'optimal', given that in previous work, the author also claimed to have identified 'optimal' fusion points which they now revise (see line 417 in the revised document). Indeed they use the term 'fully optimize' in the discussion (line 459), which implies there are degrees of optimization.

We have changed our description of the junctions in the abstract from "optimal" to "optimized".

"We then probed each position in the AT linker region to determine how domain boundaries influence structural integrity and identified a new set of **optimized** domain boundaries."

2. Response to comment 6: the authors included 'a representative result from LC-MS measurement', but all of the raw data files should be made available, as is standard for the journal.

We have included a file in the Source data file that contains the LC-MS output files. We have also included data measurements of methyl TKL in Figure 5f.

2. Text written in response to comment 23: 'frequent' should be 'frequently'

The text has been changed according to the reviewer's suggestion:

"Interestingly, while the reference set showed a mostly even distribution of junctions, except for positions us80-us102 which appeared less **frequently** than expected, the high solubility set exhibited a clear bias against junctions at certain positions (Fig. 3c, d)."

3. Text written in response to comment 41: 'the linkers wrap around KS-AT linkers and continue...' The grammar here should be corrected.

The original text wrote "continue" when the correct grammar should be "continues" and "wrap" instead of "wraps". This error has been corrected:

"The AlphaFold model prediction of the post-AT linker on the other hand, wraps itself around the residues of the KS-AT linker and continues along the KS domain"

4. Text, line 382. It is too precise to state that 'around a hundred natural polyketide analogs have been generated', as to my knowledge, no one has published an exact count.

We changed the text from saying "a hundred" to now say "many":

"Although **many** natural polyketide analogs have been generated by rational protein engineering, unfortunately the resulting engineered PKSs generally have significantly reduced kinetics compared with their wild-type counterparts, producing low titers of desired molecules^{6,7}"

Reviewer #3 (Remarks to the Author):

I thank the authors for their detailed responses and edits and believe that all concerns and errors were adequately addressed in the revised manuscript. I therefore recommend the revised manuscript for publication.